# REPRESENTATION NORM AMPLIFICATION FOR OUT-OF-DISTRIBUTION DETECTION IN LONG-TAIL LEARNING

## ABSTRACT

Detecting out-of-distribution (OOD) samples is a critical task for reliable machine learning. However, this task becomes particularly challenging when the models are trained on long-tailed datasets, as the models often struggle to distinguish tail-class in-distribution samples from OOD samples. We examine the main challenges in this problem by identifying the trade-offs between OOD detection and in-distribution (ID) classification, faced by existing methods. We then introduce our method, called *Representation Norm Amplification* (RNA), which solves this challenge by decoupling the two problems. The main idea is to use the norm of the representation as a new dimension for OOD detection, and to develop a training method that generates a noticeable discrepancy in the representation norm between ID and OOD data, while not perturbing the feature learning for in-distribution classification. Our experiments show that RNA achieves superior performance in both OOD detection and classification compared to the state-of-the-art methods, by 2.36%, 1.17%, and 7.38% in AUROC and 2.20%, 0.95%, and 2.84% in classification accuracy on CIFAR10-LT, CIFAR100-LT, and ImageNet-LT, respectively.

## 1 INTRODUCTION

The issue of overconfidence in machine learning has received significant attention due to its potential to produce unreliable or even harmful decisions. One common case when the overconfidence can be harmful is when the model is presented with inputs that are outside its training distribution, also known as out-of-distribution (OOD) samples. To address this problem, OOD detection, i.e., the task of identifying inputs outside the training distribution, has become an important area of research (Nguyen et al., 2015; Bendale & Boult, 2016; Hendrycks & Gimpel, 2017). Among the notable approaches is Outlier Exposure (OE) (Hendrycks et al., 2019a), which uses an auxiliary dataset as an OOD training set and regulates the model's confidence on that dataset by regularizing cross-entropy loss (Hendrycks et al., 2019a), free energy (Liu et al., 2020), or total variance loss (Papadopoulos et al., 2021). These methods have proven effective and become the state-of-the-art in OOD detection. However, recent findings have highlighted the new challenges in OOD detection posed by long-tailed datasets, characterized by class imbalance, which are common in practice. Even OE and other existing techniques struggle to distinguish tail-class in-distribution samples from OOD samples in this scenario (Wang et al., 2022b). Thus, improved methods are needed to address this new challenge.

To address OOD detection in long-tail learning, we need to solve two challenging problems: (i) achieving high classification accuracy on balanced test datasets, and (ii) distinguishing OOD samples from both general and tail-class in-distribution (ID) data, which is underrepresented in the training set. While previous works have explored each problem separately, simply combining the long-tailed recognition (LTR) methods (Kang et al., 2020; Menon et al., 2021) with OOD methods does not effectively solve this challenge (Wang et al., 2022b). This is because the existing LTR and OOD methods often pursue conflicting goals in the logit space (network output), creating trade-offs between OOD detection and classification, especially for tail classes (see Table 1). For example, OE (Hendrycks et al., 2019a) uses OOD auxiliary samples to train the model to generate a uniform logit vector for rare samples, while the LTR method such as Logit Adjustment (LA) encourages a relatively large margin between the logits of tail versus head classes. Combining these approaches results in better OOD detection but inferior classification compared to LTR-tailored methods, while achieving worse OOD detection but better classification than OOD-tailored methods. The crucial question remains: how can we achieve both goals without compromising each other?

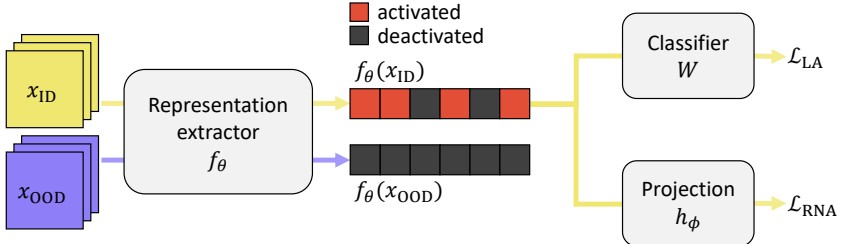

Figure 1: Overview of the proposed method, Representation Norm Amplification (RNA): During training, RNA uses both in-distribution (ID) and auxiliary OOD data. The network parameters are updated to minimize the classification loss $\mathcal{L}_{\text{LA}}$ (Eq. 5) of the ID samples, regularized by their representation norms through the RNA loss $\mathcal{L}_{\text{RNA}}$ (Eq. 3). The OOD data does not contribute to model parameter updates, but is used to update the running statistics of the Batch Normalization (BN) layers. Our proposed method effectively generates a noticeable discrepancy in the activation ratio at the last ReLU layer (before the classifier) and the representation norm between ID vs. OOD data, which serves as the basis for our OOD score, allowing for reliable OOD detection.

To address this challenge, we provide a new OOD detection method in long-tail learning, called Representation Norm Amplification (RNA), which can effectively decouple the classification and OOD detection problems, by performing the classification in the logit space and the OOD detection in the embedding space. Our proposed training method creates a distinction between ID and OOD samples based on the norm of their representations, which serves as our OOD scoring method. The main idea is to train the classifier to minimize classification loss only for ID samples, regularized by a loss enlarging the norm of ID representations. Meanwhile, we pass auxiliary OOD data through the network to regularize the Batch Normalization (BN) (Ioffe & Szegedy, 2015) layers using both ID and OOD data (Fig. 1). This simple yet effective training method generates a discernible difference in the activation ratio at the last ReLU layer (before the classifier) and the representation norm between ID vs. OOD data, which proves valuable for OOD detection. In addition, since the loss only involves ID data, only ID data contributes to the gradient for updating model parameters. This focus on learning representations for classification, rather than controlling network output for OOD data, is a key aspect in achieving high classification accuracy. Our contributions can be summarized as follows:

- We introduce Representation Norm Amplification (RNA), a novel training method that successfully disentangles classification and OOD detection in long-tail learning. RNA achieves this by intentionally inducing a noticeable difference in the representation norm between ID and OOD data. This approach enables effective OOD detection without compromising the models' capability to learn representations for achieving high classification accuracy.
- We evaluate RNA on diverse OOD detection benchmarks and show that it improves AUROC by 2.36%, 1.17%, and 7.38% and classification accuracy by 2.20%, 0.95%, and 2.84% on CIFAR10/100-LT and ImageNet-LT, respectively, compared to the state-of-the-art method.

## 2 MOTIVATION: TRADE-OFFS BETWEEN OOD DETECTION AND LONG-TAILED RECOGNITION

We first examine the OOD detection in long-tailed recognition to understand the main challenges. Specifically, we observe trade-offs between the two problems, when the state-of-the-art OOD detection method, Outlier Exposure (OE) (Hendrycks et al., 2019a), is simply combined with the widely used long-tail learning method, Logit Adjustment (LA) (Menon et al., 2021). OE exposes auxiliary OOD samples during training to discourage confident predictions on rare OOD samples by enforcing a uniform logit for OOD samples, while LA encourages confident predictions on rare tail-class in-distribution samples by applying a label frequency-dependent offset to each logit during training.

In Table 1, we compare the performance of models trained with the cross-entropy (CE) loss, CE+OE losses (OE), LA, LA+OE, and PASCL (Wang et al., 2022b), which is a recently developed method for OOD detection in long-tailed recognition by using the contrastive (contr.) learning idea, combined with LA+OE. We report the results of ResNet18 (He et al., 2015) trained with CIFAR10-LT (Cui

Table 1: The existing OOD detection method, Outlier Exposure (OE), and the long-tailed recognition method, logit adjustment (LA), aim to achieve better OOD detection and classification, resp., by enforcing the desired confidence gaps between in-distribution vs. OOD or head vs. tail classes. Simply combining OE and LA (LA+OE) leads to trade-offs between OOD detection performance (FPR95) and classification accuracy, especially for the tail classes (Few). Our method (RNA) achieves superior OOD detection and classification performance by decoupling the two problems.

| Method | Softmax Confidence | | | FPR95 ($\downarrow$) | | Accuracy ($\uparrow$) | |
|---|---|---|---|---|---|---|---|
| | Avg. | Few | OOD | Avg. | Few | Avg. | Few |
| CE | 93.51 | 90.16 | 87.03 | 59.78 | 80.17 | 72.65 | 51.32 |
| CE+OE | 65.04 | 46.64 | 10.91 | 16.49 | 25.64 | 74.85 | 56.79 |
| LA | 92.75 | 89.54 | 81.65 | 52.64 | 66.04 | 77.30 | 64.18 |
| LA+OE | 64.01 | 46.63 | 10.94 | 17.92 | 28.45 | 75.32 | 57.93 |
| LA+OE+contr. (PASCL) | 65.58 | 48.18 | 10.91 | 13.30 | 19.28 | 76.39 | 65.72 |
| RNA (ours) | 84.23 | 75.93 | 13.33 | **10.13** | **16.69** | **78.59** | **68.25** |

et al., 2019), when the performance is evaluated for an OOD test set, SVHN (Netzer et al., 2011). FPR95 is an OOD detection score, indicating the false positive rate (FPR) for ID data at a threshold achieving the true positive rate (TPR) of 95%. FPR95 can be computed for each class separately, and we report the FPR95 scores averaged over all classes (Avg.) and the bottom 33% of classes (Few) (in terms of the number of samples in the training set), and similarly for the classification accuracies.

Comparing the results between CE and CE+OE and between LA and LA+OE, we can observe that OE increases the gap between the softmax confidence scores of ID and OOD data, which makes it easier to distinguish ID data from OOD data by the confidence score, resulting in the lower FPR95 values. However, compared to the model trained with LA, the model trained with LA+OE achieves a lower classification accuracy, reduced by 1.98% on average and 6.25% for the 'Few' class group.

One can view LA+OE as a method that trades off the OOD detection performance and the classification accuracy, especially for tail classes, since it achieves worse FPR95 but better classification accuracy compared to the OOD-tailored method, CE+OE, while it achieves better FPR95 but worse accuracy than the LTR-tailored method, LA. PASCL, which uses contrastive learning to push the representations of the tail-class in-distribution samples and the OOD auxiliary samples, improves both the FPR95 and the classification accuracy for the 'Few' class group, each compared to those of CE+OE and LA, respectively. However, it still achieves a slightly lower average classification accuracy (-0.91%) than the model trained only for long-tailed recognition by LA loss.

The main question is then whether such trade-offs between OOD detection and long-tailed recognition are fundamental, or whether we can achieve the two different goals simultaneously, without particularly compromising each other. All the existing methods reviewed in Table 1 aim to achieve better OOD detection, long-tailed recognition, or both by enforcing the desired logit distributions (and the corresponding confidence levels) for ID vs. OOD data or head vs. tail classes, respectively. However, we observe the limitations of such approaches in achieving the two possibly conflicting goals simultaneously through the logit space, even with the help of supervision by contrastive learning.

Based on this observation, we propose to utilize a new dimension for OOD detection, *the norm of representation vectors*, to decouple the OOD detection problem and the long-tailed recognition problem, and to achieve the two different goals without perturbing each other. As shown in Table 1, our method, Representation Norm Amplification (RNA), achieves the best FPR95 and classification accuracy for both the average case and the 'Few' class group, compared to other methods, including even those tailored to only one of the two goals, such as CE+OE or LA. In particular, our method achieves a desirable gap between the confidence scores of ID (Avg.) vs. OOD data (84.23 vs. 13.33%), while maintaining a relatively high confidence score of 75.93%, for the 'Few' ID classes.

## 3 METHOD: REPRESENTATION NORM AMPLIFICATION (RNA)

In Sec. 3.1, we discuss the limitations of previous approaches in the way of exposing OOD samples during training by analyzing the loss gradients, and in Sec. 3.2, we present our method.

### 3.1 PREVIOUS WAYS OF EXPOSING AUXILIARY OOD DATA DURING TRAINING

Let $\mathcal{D}_{\text{ID}}$ and $\mathcal{D}_{\text{OOD}}$ denote an ID training dataset and an auxiliary OOD dataset, respectively. The state-of-the-art OOD detection methods expose the auxiliary OOD data during training to minimize the combination of a classification loss $\mathcal{L}_{\text{ID}}$ for ID data and an extra loss $\mathcal{L}_{\text{OOD}}$ for OOD data, i.e., $\mathcal{L} = \mathcal{L}_{\text{ID}} + \lambda\mathcal{L}_{\text{OOD}}$ for some tunable parameter $\lambda > 0$. As an example, one can choose the regular softmax cross-entropy loss as the ID loss, $\mathcal{L}_{\text{ID}} = \mathbb{E}_{(x,y)\sim\mathcal{D}_{\text{ID}}}\left[\log\left[\sum_{c\in[C]}\exp(w_c^\top f_\theta(x))\right] - (w_y^\top f_\theta(x))\right]$, where $f_\theta(x) \in \mathbb{R}^D$ is the representation of the input data $x$ and $w_c \in \mathbb{R}^D$ is the classification weight (i.e., the last-layer weight) for the label $c \in [C] := \{1, \ldots, C\}$. For long-tailed recognition, the LA method (Menon et al., 2021) uses the logit-adjusted loss to improve the classification accuracy of ID data, i.e., $\mathcal{L}_{\text{ID}} = \mathcal{L}_{\text{LA}}$ in Eq. 5. On the other hand, the loss for OOD training data is often designed to control the logit distribution or the softmax confidence of the OOD samples. For example, the OE method minimizes the cross entropy between the uniform distribution over $C$ classes and the softmax of the OOD sample, i.e., $\mathcal{L}_{\text{OOD}} = \mathbb{E}_{x'\sim\mathcal{D}_{\text{OOD}}}\left[\log\left[\sum_{c\in[C]}\exp(w_c^\top f_\theta(x'))\right] - \sum_{c\in[C]}\frac{1}{C}\left(w_c^\top f_\theta(x')\right)\right]$.

**OE perturbs tail classification: gradient analysis** We explain why training with OOD data to minimize the loss of the form $\mathcal{L} = \mathcal{L}_{\text{ID}} + \lambda\mathcal{L}_{\text{OOD}}$ can lead to a degradation in classification accuracy, particularly for tail classes. Let $\mathcal{T} = \{(x_i, y_i), x_i'\}_{i=1}^{B}$ be a training batch sampled from the sets of ID training samples $\{(x_i, y_i)\}$ and unlabeled OOD samples $\{x_i'\}$, respectively. For simpleness of analysis, we assume a fixed feature extractor $f_\theta(\cdot)$ and examine how ID/OOD samples in the batch affect the updates of the classification weights $\{w_c\}_{c=1}^{C}$. Note that the gradient of the loss $\mathcal{L} = \mathcal{L}_{\text{ID}} + \lambda\mathcal{L}_{\text{OOD}}$ on the batch $\mathcal{T}$ with respect to the classification weight $w_c$ is

$$\frac{\partial\mathcal{L}}{\partial w_c} = \frac{1}{B}\sum_{i=1}^{B} f_\theta(x_i)\left(\mathcal{S}(W^\top f_\theta(x_i))_c - y_i^c\right) + \frac{\lambda}{B}\sum_{i=1}^{B} f_\theta(x_i')\left(\mathcal{S}(W^\top f_\theta(x_i'))_c - 1/C\right), \quad (1)$$

where $\mathcal{S}(W^\top f_\theta(x_i))_c$ is the softmax output of label $c$ for the weight matrix $W = [w_1, w_2, \ldots, w_C]$, and $y_i^c = 1$ if $y_i = c$ and 0 otherwise. For $\mathcal{L}_{\text{ID}} = \mathcal{L}_{\text{LA}}$, $\mathcal{S}(W^\top f_\theta(x_i))_c$ in the first term is replaced by the softmax output for the logits adjusted by the label-dependent offsets as in Eq. 5.

From Eq. 1, the gradient is a weighted sum of the representations of ID data $\{f_\theta(x_i)\}_{i=1}^{B}$ and OOD data $\{f_\theta(x_i')\}_{i=1}^{B}$, where the weights depend on the gap between the softmax output $\mathcal{S}(W^\top f_\theta(x_i))_c$ and the desired label, $y_i^c$ for ID data and $1/C$ for OOD data, respectively. Thus, the gradient balances the effects of ID samples, tuned to minimize the classification error, and those of OOD samples, tuned to control their confidence levels. For a tail class $t$, however, the proportion of ID samples with the label $y_i = t$ is relatively small. As a result, during training, the gradient for the classification weight $w_t$ of the tail class is dominated by the representations of OOD samples rather than those of the tail-class ID samples, particularly at the early stage of training when the label distribution for OOD samples deviates significantly from a uniform distribution. In Figure 2a, we plot the ratios $(\|\nabla_{w_c}\mathcal{L}_{\text{ID}}\|_1 / \|\nabla_{w_c}\mathcal{L}_{\text{OOD}}\|_1)$ of the gradient norms between ID and OOD samples with respect to the classification weights of bottom 3 (tail) classes vs. top 3 (head) classes. Classification weights for tail classes are more heavily influenced by OOD samples than those for head classes, especially in the early stages of training, resulting in a greater degradation of classification accuracy, as observed in Table 1. The question is then how to effectively expose OOD data during training to learn the representations to distinguish ID and OOD data, while not degrading the classification.

### 3.2 PROPOSED OOD SCORING AND TRAINING METHOD WITH AUXILIARY OOD DATA

**Representation Norm (RN) score** We present a simple yet effective approach to expose OOD data during training to create a discrepancy in the representation between ID and OOD data. The main idea is to utilize the norm of representations (i.e., feature vectors). Note that the representation vector $f_\theta(x)$ can be decomposed into its norm $\|f_\theta(x)\|_2$ and direction $\hat{f}_\theta(x) = f_\theta(x)/\|f_\theta(x)\|_2$ as $f_\theta(x) = \|f_\theta(x)\|_2\hat{f}_\theta(x)$. Simply scaling the norm $\|f_\theta(x)\|_2$ of the representation without changing its direction does not change the class prediction, since for any scaling factor $s > 1$, we have $\arg\max_{c\in[C]}\mathcal{S}(W^\top s f_\theta(x))_c = \arg\max_{c\in[C]}\mathcal{S}(W^\top f_\theta(x))_c$. We thus use this new dimension, the norm of representations, to generate a discrepancy between the representations of ID vs. OOD data, while not enforcing the desired logit distribution for OOD data. Furthermore, we define a new OOD

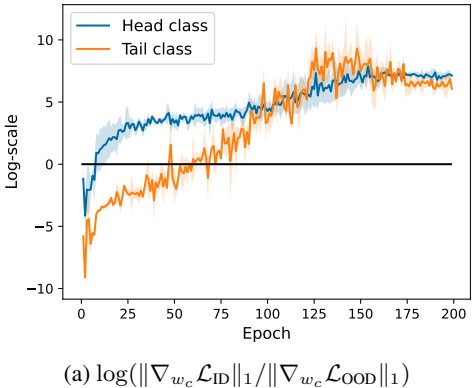 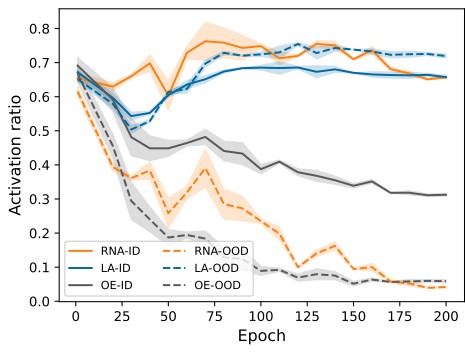

(a) $\log(\|\nabla_{w_c}\mathcal{L}_{\mathrm{ID}}\|_1/\|\nabla_{w_c}\mathcal{L}_{\mathrm{OOD}}\|_1)$       (b) The activation ratio at the last ReLU layer

Figure 2: (a) The gradient ratio of ID classification loss to OOD detection loss with respect to the classifier weight of LA+OE model trained on CIFAR10-LT. In particular, $\mathcal{L}_{\mathrm{LA}}$ and $\mathcal{L}_{\mathrm{OE}}$ are used as $\mathcal{L}_{\mathrm{ID}}$ and $\mathcal{L}_{\mathrm{OOD}}$, respectively. Note that the log-ratio for tail classes is less than zero at the early stage of training, indicating that the gradient update is dominated by OOD data rather than ID data. (b) The activation ratio of ID and OOD representations at the last ReLU layer in the models trained by RNA, LA, and OE. CIFAR10 and SVHN are used as ID and OOD sets, respectively.

scoring method, called *Representation Norm (RN)* score, using the norms of representations as

$$S_{\mathrm{RN}}(x) = -\|f_\theta(x)\|_2, \tag{2}$$

for any test sample $x$. Following the convention for OOD detection, samples with larger scores are detected as OOD data, while those with smaller scores are considered as ID data.

**Representation Norm Amplification (RNA)**    We propose a training method to obtain desired representations for ID vs. OOD data. Our goal in the representation learning is twofold: (i) to make the norms of the representations for ID data relatively larger than those of OOD data, and (ii) to learn the representations that achieve high classification accuracy even when trained on long-tailed datasets. To achieve the first goal, we propose Representation Norm Amplification (RNA) loss $\mathcal{L}_{\mathrm{RNA}}$:

$$\mathcal{L}_{\mathrm{RNA}} = \mathbb{E}_{x\sim\mathcal{D}_{\mathrm{ID}}}\left[-\log(1 + \|h_\phi(f_\theta(x))\|)\right], \tag{3}$$

where $h_\phi$ is a 2-layer MLP projection function. This loss is designed to increase the norm of the representations for ID data. We then combine this loss with the logit-adjusted (LA) loss $\mathcal{L}_{\mathrm{LA}}$ (Menon et al., 2021), tailored for long-tail learning, and define our final loss function as below:

$$\mathcal{L} = \mathcal{L}_{\mathrm{LA}} + \lambda\mathcal{L}_{\mathrm{RNA}}, \tag{4}$$

where $\lambda > 0$ is a hyperparameter balancing the two objectives. We set $\lambda = 0.5$ for all the experiments in Sec. 4. The LA loss (the first term) is defined as:

$$\mathcal{L}_{\mathrm{LA}} = \mathbb{E}_{(x,y)\sim\mathcal{D}_{\mathrm{ID}}}[\log[\sum_{c\in[C]}\exp\left(w_c^\top f_\theta(x) + \tau\log\pi_c\right)] - (w_y^\top f_\theta(x) + \tau\log\pi_y)], \tag{5}$$

where $\tau > 0$ is a hyperparameter, which we set to 1, and $\pi \in [0,1]^C$ is the estimate of the label distribution over $C$ classes in the training dataset.

It is important to note that we only use in-distribution data to optimize the loss function (Eq. 4) by gradient descent. The OOD training data does not provide a gradient to update the model parameters; instead, it is simply fed into the model as shown in Fig. 1. This limits its influence on learning the network parameters. However, we use OOD data to regularize the model by updating the running statistics of the BN layers from both in-distribution and OOD data. In the rest of this section, we will explain why forwarding OOD data is sufficient to regularize the model to produce relatively smaller representation norms of OOD data than those of ID data, when the total loss is regularized by $\mathcal{L}_{\mathrm{RNA}}$.

**Effect of auxiliary OOD data: regularizing the activation ratio**    In commonly used deep neural network architectures, BN layers are typically followed by ReLU layers (Nair & Hinton, 2010),

where the BN layers standardize inputs by estimating the mean and variance of all the given data. Consequently, the coordinates of input vectors with values below the running mean of the BN layer are deactivated as they pass through the next ReLU layer. In Fig. 2b, we compare the activation ratio (the fraction of activated coordinates) at the last ReLU layer (prior to the last linear layer) for ID data, CIFAR-10, and OOD data, SVHN, using models trained with RNA, LA, and OE. Our method (RNA) exhibits discernible patterns in the activation ratio, clearly distinguishing between ID and OOD data. RNA activates the majority of the coordinates of representation vectors of ID data (66.08% on CIFAR10) while deactivating those of OOD data (4.17% on SVHN). This illustrates that regularizing the BN layers using both ID and OOD data, while enlarging the ID representation norms through $\mathcal{L}_{\text{RNA}}$, results in a substantial difference between the activation ratio at the last ReLU layer. Additionally, since the activation ratio at the last ReLU layer corresponds to the fraction of non-zero entries in the representation vector, a significant gap in the activation ratio leads to a difference in the norm of representations between ID and OOD data, allowing for reliable OOD detection by the RN score (Eq. 2). The model trained with LA, which aims to increase the confidence of 'rare' (tail-class) samples, exhibits high activation ratios for both ID and OOD samples (65.75% on ID and 71.93% on OOD). In contrast, the model trained with OE method has a low activation ratio for both ID and OOD samples (31.25% and 7.92%, respectively) while still generating a considerable gap between the two groups, despite not including any regularization loss for representation norm. This gap in the activation ratio between ID and OOD data enables our RN score to detect OOD samples effectively, even in the models trained by OE, as will be demonstrated in our ablation study (Table 3).

## 4 EXPERIMENTAL RESULTS

### 4.1 EXPERIMENTAL SETUP

**Datasets and training setup** For in-distribution datasets, we use CIFAR10/100 (Krizhevsky, 2009) and ImageNet-1k (Deng et al., 2009). The long-tailed training sets, CIFAR10/100-LT, are built by downsampling CIFAR10/100 (Cui et al., 2019), making the imbalance ratio, $\max_c N(c)/\min_c N(c)$, equal to 100, where $N(c)$ is the number of samples in class $c$. The results for more diverse imbalance ratios are reported in Appendix E.3. We use 300K random images (Hendrycks et al., 2019a) as an auxiliary OOD training set for CIFAR10/100. For ImageNet, we use ImageNet-LT (Liu et al., 2019) as the long-tailed training set and ImageNet-Extra (Wang et al., 2022b) as the auxiliary OOD set.

For experiments on CIFAR10/100, we use the semantically coherent out-of-distribution (SC-OOD) benchmark datasets (Yang et al., 2021) as our OOD test set. For CIFAR10 (respectively, CIFAR100), we use Textures (Cimpoi et al., 2014), SVHN (Netzer et al., 2011), CIFAR100 (respectively, CIFAR10), Tiny ImageNet (Le & Yang, 2015), LSUN (Yu et al., 2015), and Places365 (Zhou et al., 2018) from the SC-OOD benchmark datasets. For ImageNet, we use ImageNet-1k-OOD, published in Wang et al. (2022b), as our OOD test set. More details on datasets can be found in Appendix D.3. We train ResNet18 (He et al., 2015) on CIFAR10/100-LT datasets for 200 epochs with a batch size of 256, and ResNet50 (He et al., 2015) on ImageNet-LT for 100 epochs with a batch size of 256. We set $\lambda = 0.5$ in (4). Further details on the implementation are available in Appendix D.1.

**Evaluation metrics** To evaluate OOD detection, we use three metrics–the area under the receiver operating characteristic curve (AUROC), the area under the precision-recall curve (AUPR), and the false positive rate of ID samples at the threshold of true positive rate of 95% (FPR95). For in-distribution classification, we measure the total accuracy and also the accuracy averaged for a different subset of classes categorized as Many, Medium, and Few in terms of the number of training samples. All the reported numbers are the average of six runs from different random seeds.

### 4.2 RESULTS

**Results on CIFAR10/100** The results on CIFAR10/100-LT are summarized in Table 2, where the OOD detection performances averaged over six different OOD test sets are reported. As baselines, we use five different OOD detection methods, including MSP (Hendrycks & Gimpel, 2017), OECC (Papadopoulos et al., 2021), EngeryOE (Liu et al., 2020), OE (Hendrycks et al., 2019a), and a recently published method specially tailored for long-tail learning, PASCL (Wang et al., 2022b). We can observe that our proposed method outperforms the baseline methods for both OOD detection and ID

Table 2: OOD detection and ID classification performance (%) on CIFAR10/100-LT and ImageNet-LT. OOD detection performance measures are reported on the left side, while classification performance measures are reported on the right side. Means and standard deviations are reported based on six runs. **Bold** indicates the best performance, and underline indicates the second best performance.

| Method | AUROC (↑) | AUPR (↑) | FPR95 (↓) | ACC (↑) | Many (↑) | Medium (↑) | Few (↑) |
|---|---|---|---|---|---|---|---|
| ID Dataset: CIFAR10-LT | | | | | | | |
| MSP | 72.68±1.04 | 70.48±0.89 | 65.67±1.82 | 72.65±0.19 | **94.65±0.27** | 72.19±0.49 | 51.32±0.74 |
| OECC | 88.08±0.19 | 88.65±0.09 | 48.48±0.88 | 74.48±0.19 | 94.29±0.29 | 72.69±0.68 | 56.79±0.68 |
| EnergyOE | 88.78±0.48 | 88.91±0.61 | 44.48±0.78 | 76.25±0.61 | 93.94±0.51 | 74.37±0.34 | 61.01±1.87 |
| OE | 89.69±0.51 | 86.47±1.41 | 33.81±0.51 | 74.85±0.57 | 93.90±0.35 | 73.30±1.12 | 57.60±1.78 |
| PASCL | 90.58±0.29 | 88.44±0.54 | 33.79±0.90 | - | - | - | - |
| PASCL (fine-tune) | - | - | - | 76.39±0.56 | 91.59±0.99 | 72.95±1.09 | 65.72±2.78 |
| RNA (ours) | **92.95±0.13** | **92.01±0.21** | **28.76±0.53** | **78.59±0.25** | 93.43±0.38 | **74.92 ±0.71** | **68.25±0.38** |
| ID Dataset: CIFAR100-LT | | | | | | | |
| MSP | 61.39±0.40 | 57.86±0.30 | 82.17±0.38 | 40.61±0.20 | **71.02±0.67** | 38.59±0.45 | 11.06±0.28 |
| OECC | 69.64±0.47 | 66.55±0.40 | 76.74±0.58 | 41.52±0.37 | 69.88±0.31 | 39.43±0.78 | 13.97±0.45 |
| EnergyOE | 69.14±0.53 | 66.67±0.21 | 80.04±1.13 | 39.17±0.34 | 68.53±0.52 | 39.54±0.47 | 8.03±0.82 |
| OE | 73.37±0.22 | 67.26±0.41 | 67.83±0.82 | 39.83±0.32 | 68.28±0.63 | 37.11±0.45 | 12.89±0.49 |
| PASCL | 73.14±0.37 | 66.77±0.50 | **67.36±0.46** | - | - | - | - |
| PASCL (fine-tune) | - | - | - | 43.44±0.34 | 65.83±0.38 | 40.76±0.60 | **22.70±0.94** |
| RNA (ours) | **74.33±0.53** | **70.25±0.49** | 68.26±0.94 | **44.39±0.14** | 67.90±0.32 | **44.24±0.62** | 20.00±0.91 |
| ID Dataset: ImageNet-LT | | | | | | | |
| MSP | 54.22±0.66 | 51.85±0.54 | 89.67±0.44 | 41.69±1.45 | 59.01±1.90 | 35.54±2.49 | 14.29±5.15 |
| OECC | 62.82±0.28 | 63.89±0.33 | 87.83±0.20 | 41.03±0.23 | 59.36±0.36 | 34.32±0.25 | 12.78±0.34 |
| EnergyOE | 63.38±0.20 | 64.51±0.23 | 88.34±0.19 | 38.47±1.00 | 58.82±0.99 | 30.61±1.17 | 8.56±0.52 |
| OE | 67.12±0.35 | 69.20±0.34 | 87.65±0.20 | 40.87±0.32 | **60.20±0.26** | 34.07±0.42 | 10.11±0.65 |
| PASCL | 68.17±0.24 | 70.26±0.31 | 87.62±0.32 | - | - | - | - |
| PASCL (fine-tune) | - | - | - | 44.97±1.01 | 55.98±1.08 | 42.29±1.06 | 23.26±0.84 |
| RNA (ours) | **75.55±0.23** | **74.60±0.27** | **78.16±0.81** | **47.81±0.31** | 58.59±0.38 | **44.58±0.38** | **28.67±0.31** |

classification. In particular, for the CIFAR10-LT dataset, RNA improves AUROC, AUPR, and FPR95 by 2.37%, 3.10%, and 5.03%, respectively, compared to the state-of-the-art method, PASCL. RNA also shows improvements of 2.20% and 2.53% over the total accuracy and accuracy for the 'Few' class group, respectively, compared to the second best model. When trained on CIFAR100-LT, RNA achieves superior AUROC and AUPR with a difference of 1.19% and 3.48%, respectively, compared to PASCL. Our method also achieves the highest overall classification accuracy of 44.39%. The results of OOD detection performance on each OOD test set are reported in Appendix E.1.

**Results on ImageNet** Table 2 also shows the results on ImageNet-LT, which is a large scale dataset. Our method significantly improves the OOD detection performance by 7.38%, 4.34%, and 9.46% for AUROC, AUPR, and FPR95, respectively. Furthermore, RNA outperforms PASCL in terms of the classification accuracy, increasing by 2.84% and 5.41% for total and 'Few' cases, respectively. It is noteworthy that the performance improvement of our method on ImageNet-LT is greater than those on CIFAR10-LT and CIFAR100-LT, indicating that our method demonstrates effective performance on large scale datasets, which is considered as a more challenging case for OOD detection.

## 4.3 ABLATION STUDY

We perform a series of ablation study to investigate our method from multiple perspectives. Our evaluations involve diverse combinations of OOD training and scoring methods, RNA-trained models with different configurations of auxiliary OOD sets, and calibration performance. Further ablation studies are available in Appendix E, where we include the results about the robustness to $\lambda$ in Eq. 4 (§E.2), different imbalance ratios (§E.3), variants of RNA training loss (§E.4), alternative auxiliary OOD sets (§E.5), the structure of the projection function $h_\phi$ (§E.6), and fine-tuning task (§E.7).

**Scoring methods** To evaluate the effect of our training method (Eq. 4) and scoring method (Eq. 2) separately, we perform an ablation study on different combinations of training losses and detection scores. In Table 3, we report the mean AUROC over six OOD test sets on models trained with

Table 3: AUROC (%) of various combinations of training methods and OOD scores on CIFAR10/100-LT and ImageNet-LT. The gray regions indicate the results for our proposed methods. MSP and Energy are confidence-based OOD scoring methods, while RN is evaluated in the embedding space.

| Training Method | OOD Scoring Method | | | | | | | | |
|---|---|---|---|---|---|---|---|---|---|
| | CIFAR10-LT | | | CIFAR100-LT | | | ImageNet-LT | | |
| | MSP | Energy | RN (ours) | MSP | Energy | RN (ours) | MSP | Energy | RN (ours) |
| CE | 72.68 | 72.77 | 70.98 | 61.39 | 61.55 | 55.67 | 54.22 | 54.07 | 56.73 |
| OE | 89.69 | 89.66 | 86.63 | 73.37 | 72.98 | 71.81 | 67.12 | 67.77 | 77.99 |
| PASCL | 90.58 | 90.71 | 81.26 | 73.14 | 72.49 | 70.68 | 68.17 | 68.70 | 77.98 |
| RNA (ours) | 92.01 | 92.29 | 92.95 | 73.93 | 74.00 | 74.33 | 61.27 | 61.50 | 75.55 |

Table 4: OOD detection and ID clsasification performance (%) on CIFAR10-LT with different usage settings of auxiliary set.

| Method | Auxiliary set | | AUROC | ACC |
|---|---|---|---|---|
| | Aux. OOD | Aug. ID | | |
| MSP | | | 72.68 | 72.65 |
| RNA | | | 75.57 | 79.32 |
| RNA | | ✓ | 83.57 | 78.13 |
| RNA | ✓ | | 92.95 | 78.59 |

Table 5: Calibration performance measured by Expected Calibration Error (ECE) (%) on CIFAR10, CIFAR100, and ImageNet.

| Method | CIFAR10 | CIFAR100 | ImageNet |
|---|---|---|---|
| CE | 20.76±0.27 | 32.06±0.25 | 21.55±1.86 |
| OE | 17.56±0.61 | **16.53±0.33** | 16.83±0.41 |
| PASCL | 18.71±0.44 | 17.17±0.25 | 16.37±0.48 |
| RNA (ours) | **12.14±0.23** | 17.12±0.53 | **12.65±0.42** |

four distinct objectives: CE, OE, PASCL and RNA, when OOD detection is performed with three different OOD scoring methods: MSP, Energy, and RN. For CIFAR10/100-LT, the model trained by RNA achieves the highest performance for all the scoring methods. This can be attributed to the fact that training with RNA indirectly increases the confidence of ID data by amplifying representation norms of ID training data. Consequently, even when MSP or Energy scores are used, which use the confidence level for OOD detection, the RNA-trained model achieves the best AUROC performance. Another interesting result is that our scoring method (RN) achieves comparable performance to MSP or Energy scoring on OE-trained models. The model trained by OE only regularizes the confidence of auxiliary OOD samples, but it eventually generates a significant gap in representation norms between ID vs. OOD data (as shown in Fig. 2b), which allows our RN score to detect OOD samples.

The experiments with ImageNet-LT show different trends. When the RNA-trained model is evaluated using confidence-based OOD scores such as MSP or Energy, it exhibits a lower AUROC compared to the models trained using OE or PASCL. However, when RN scoring is used, all the training methods (OE, PASCL and RNA) achieve significant improvements of 10.22%, 9.28%, and 14.05% over Energy scoring, respectively. The different trends may be due to the difference in the number of classes of each dataset, 10/100 for CIFAR10/100 and 1,000 for ImageNet-1k. The models trained on ImageNet-LT generally produce low confidence scores due to the large number of classes. Thus, the confidence-based score may have a higher chance of incorrectly classifying ID samples as OOD samples, reducing the AUROC performance. This highlights the superiority of the representation-based OOD scoring method (RN), especially when dealing with a large number of classes.

**Use of auxiliary OOD data**    We conduct an ablation study to assess the significance of incorporating auxiliary OOD data in our method. The results, as presented in Table 4, reveal that employing the RNA loss without any auxiliary data leads to a modest AUROC improvement of merely 2.89% over the baseline MSP approach. In contrast, the combination of RNA loss with BN regularization, involving auxiliary OOD samples, yields a substantial gain of 20.27%. This clearly highlights the significant role played by BN regularization using auxiliary OOD data, coupled with the RNA loss.

We also explore the potential for eliminating the requirement of auxiliary samples. We conduct an experiment wherein the auxiliary OOD dataset is substituted with augmented images derived from the original training samples. We employ an augmentation strategy involving random cropping that spans 5% to 25% of the original image size. This strategy achieves a notable AUROC performance gain of 10.89% over the MSP baseline, and an 8.00% gain over the case of not utilizing any auxiliary samples but applying RNA loss. Although this outcome still falls short of matching the performance gains achieved by our original method with given the auxiliary OOD set, it indicates the potential for replacing auxiliary OOD samples with augmented training samples in applying our method.

**Calibration** We evaluate the calibration performance of the models trained by our method on CIFAR10/100-LT and ImageNet-LT. To measure calibration, we utilize the Expected Calibration Error (ECE), a metric defined as $\text{ECE} := \sum_{m=1}^{M} \frac{|B_m|}{n} |\text{ACC}(B_m) - \text{Conf}(B_m)|$, where $B_m := \{x : (m-1)/M < \text{Conf}(x) \leq m/M\}$ represents the confidence bin, $\text{ACC}(B_m)$ and $\text{Conf}(B_m)$ denote the average accuracy and confidence within the bin $B_m$, respectively, and $n$ is the total number of data in the test set. In Table 5, we report the measured ECE values for models trained with CE, OE, PASCL, and RNA losses. The models are trained on the long-tailed datasets, and the ECE values are evaluated on the balanced test sets. Our proposed method, RNA, achieves the best ECE values for CIFAR10 and ImageNet, and the second-best ECE value for CIFAR100. This result demonstrates that our method exhibits strong calibration performance for ID data.

## 5 RELATED WORKS

**OOD detection in long-tail learning** While substantial research has been conducted in OOD detection and long-tail learning independently, the investigation of their combined task, referred to as LT-OOD, has only recently gained attention. PASCL (Wang et al., 2022b) first highlighted the challenge of OOD detection within the context of long-tail learning and introduced the contrastive learning idea to improve separability between ID tail classes and OOD data. However, it requires a separate model fine-tuned solely for long-tail learning to achieve both high classification accuracy and OOD detection performance. Choi et al. (2023) recently addressed LT-OOD task, highlighting that a general imbalance across classes exists in the distribution of auxiliary OOD dataset. They proposed a method to regularize auxiliary samples from majority classes, more heavily than those from minority classes, when applying the energy-based regularization for auxiliary OOD samples. To further boost ID classification accuracy, similar to (Wang et al., 2022b), this paper uses auxiliary branch fine-tuning technique. In contrast, our method does not use such two-branch technique, each tailored for OOD detection and ID classification. Instead, we address both challenges through a single model, by leveraging the idea to disentangle classification and OOD detection, allowing for effective OOD detection and classification, as observed in Sec. 4.2. Jiang et al. (2023) also tackled LT-OOD task using a post-hoc approach by adapting traditional OOD scoring methods to the long-tailed setting. While this method proves to be effective, its success is attributed to the inherent bias in the model induced by the class imbalance in the training dataset. Thus, it is hard to be combined with long-tail learning methods that resolve the class imbalance issue to improve the classification accuracy.

**Controlling norm of logits/representations** There are several previous works that have explored techniques for adjusting/normalizing the norm of logits or representations to improve OOD detection performance. For example, Objectosphere (Dhamija et al., 2018) directly attenuates the representation norm of auxiliary OOD data during training to increase separation in deep feature space between ID and OOD data. However, as detailed in Appendix E.4, training with RNA loss is more effective than directly attenuating OOD representation norms to reduce activation ratios and representation norms. There also exist techniques to mitigate overconfident issue for ID data. ODIN (Liang et al., 2018) employs temperature scaling for softmax scores at test time to mitigate overconfident predicitions, while LogitNorm (Wei et al., 2022) normalizes logits of all samples during training. However, these techniques do not take into account the underconfident issue of tail classes. In long-tail learning, the networks often produce underconfident outputs for tail-class predictions. Thus, we need to solve both the overconfidence problem of OOD samples and the underconfidence problem of tail-class ID samples. In our method, we address this challenge by amplifying the norms of only ID representations, while indirectly reducing the representation norms of OOD data by updating the BN statistics using both ID and OOD data. More reviews of related works are available in Appendix A.

## 6 DISCUSSION

We propose a simple yet effective method for OOD detection in long-tail learning, Representation Norm Amplification (RNA). Our method regularizes the representation norm of only ID data, while updating the BN statistics using both ID and OOD data, which effectively generates a noticeable discrepancy in the representation norm between ID and OOD data. The effectiveness of our method is demonstrated with experimental results on various OOD test sets and long-tailed training sets. Broader impact, limitations and further discussion of our work are available in Appendix B.

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

# A    DETAILED REVIEW ON RELATED WORKS

In this section, we provide more detailed review on some of the related works, discussed in Sec. 5.

**OOD detection**    OOD scores are typically evaluated based on output confidence (Bendale & Boult, 2016; Hendrycks & Gimpel, 2017; Liang et al., 2018; Liu et al., 2020), feature space properties (Lee et al., 2018b; Dong et al., 2021; Sun et al., 2021; Song et al., 2022; Sun & Li, 2022; Sun et al., 2022; Vaze et al., 2022; Wang et al., 2022a; Ahn et al., 2023; Djurisic et al., 2023; Yu et al., 2023; Zhu et al., 2022b), or gradients (Huang et al., 2021). Many methods have been developed to encourage the separability of ID and OOD data by these scores, for example by exposing auxiliary OOD data during training and regularizing their confidence levels (Malinin & Gales, 2018; Hein et al., 2019; Liu et al., 2020; Mohseni et al., 2020; Papadopoulos et al., 2021; Yang et al., 2021; Katz-Samuels et al., 2022; Ming et al., 2022), or by promoting the ID-OOD separability in the embedding space (Dhamija et al., 2018; Lee et al., 2018a; Hendrycks et al., 2019b; Choi & Chung, 2020; Hsu et al., 2020; Tack et al., 2020; Sehwag et al., 2021; Wang et al., 2021; Ming et al., 2023). However, these methods often suffer from performance degradation in a long-tail learning setup (Wang et al., 2022b).

**Long-tail learning**    Long-tail learning addresses the classification problem when trained on imbalanced or long-tailed class distributions. When trained on imbalanced datasets, the model is often biased towards the majority classes, generating underconfident predictions on the minority or tail classes. There have been many methods to fix this problem by over- or under-sampling (Kubat et al., 1997; Chawla et al., 2002; Pouyanfar et al., 2018; Kang et al., 2021), adjusting margins or decision thresholds (Cao et al., 2019; Tan et al., 2020; Menon et al., 2021), or modifying the loss functions (Menon et al., 2021; Cui et al., 2021; Li et al., 2022b; Zhu et al., 2022a). In particular, some methods attempt to increase the value of the logit for the tail classes by the post-hoc normalization of the weight of the classifier (the last linear layer) (Kang et al., 2020), or by applying a label frequency-dependent offset to each logit during training (Menon et al., 2021). While these approaches have shown empirical effectiveness and statistical robustness in the context of long-tail learning, they often conflict with the principle of out-of-distribution (OOD) detection methods, which regulate the confidence of models on rare samples to prevent overconfident predictions (Hendrycks et al., 2019a; Papadopoulos et al., 2021).

**OOD detection and controlling overconfidence**    There exist OOD detection methods that do not use auxiliary outlier data during training, but effectively mitigate the overconfidence issue of the models. For example, LogitNorm (Wei et al., 2022) characterizes that training with cross-entropy loss causes logit norms to keep increasing as the training progresses, and this often leads to overconfident predictions of the models on both in-distribution and OOD test data. Therefore, the LogitNorm method normalizes the logit of every input data during training to prevent the logit norm from increasing too much. Some recently published works, Rectified Activations (ReAct) (Sun et al., 2021) and Directed Sparisification (DICE) (Sun & Li, 2022), on the other hand, focus on the fact that on a model trained only on in-distribution data, OOD data still activate a non-negligible fraction of units in the penultimate layer, where the mean activation is biased towards having sharp positive values and there are "noisy units" with high variances of contribution to the class output. Thus, these methods use post-hoc activation truncation (Sun et al., 2021) or weight sparsification (Sun & Li, 2022), to avoid overconfident predictions on OOD data.

# B    BROADER IMPACTS AND LIMITATIONS

Mitigating OOD uncertainty is crucial, particularly in safety-critical applications such as medical diagnosis and autonomous driving. In these domains, the ability to effectively identify and handle OOD samples is essential to ensure the reliability and safety of the systems. By addressing the challenges associated with OOD detection, we can significantly enhance the trustworthiness and applicability of deep learning models in these critical domains.

Our proposed method specifically focuses on tackling the challenging scenarios where reliable OOD detection needs to be performed on models trained with long-tailed distributions, which is a common occurrence in practical settings but has received limited attention in previous research. By introducing our method, we enhance the safety and reliability of deploying deep models, even when trained on

imbalanced datasets that have not been extensively preprocessed for class balance. This enables the application of deep learning models in real-world scenarios, where long-tailed distributions are prevalent, without compromising their OOD detection capabilities.

Our proposed method does have some limitations. Firstly, it relies on the availability of an auxiliary OOD dataset for effective performance. We address this issue by investing the possibility of replacing the auxiliary OOD dataset by augmented training dataset as reported in Table 4.

Additionally, as shown in Table 6, our method demonstrates suboptimal performance on the near OOD test set, where the OOD data is semantically similar to the training data. This limitation arises because the representation distributions of the near OOD test data are closer to the in-distribution training data rather than the auxiliary OOD data, resulting in some test OOD samples having large representation norms. This vulnerability to near OOD data can be alleviated by employing confidence-based OOD scoring methods. However, it entails a trade-off in OOD detection performance for far OOD data, as discussed in Sec. E.1. To address this limitation, future research should focus on developing training methods that can enhance the OOD detection performance specifically for near OOD test sets in long-tail learning scenarios. By overcoming this challenge, we can further strengthen the robustness and practical applicability of OOD detection methods in more broad real-world settings.

## C  PSEUDOCODE

The training scheme of Representation Norm Amplification (RNA) is shown in Algorithm 1, and evaluation scheme of Representation Norm (RN) is shown in Algorithm 2.

---

**Algorithm 1:** RNA: Representation Norm Amplification

**Input :** In-distribution training set $\mathcal{D}_{\text{ID}}$, auxiliary OOD training set $\mathcal{D}_{\text{OOD}}$, batch size $b$, number of classes $C$, representation extractor $f_\theta$, linear classifier $W = [w_1, w_2, \ldots, w_C]$, projection function $h_\phi$, learning rate $\eta$

1 **for** $epoch = 1, 2, \ldots,$ **do**
2    **for** $iter = 1, 2, \ldots,$ **do**
      // Sample a batch of ID and OOD samples.
3       $\mathcal{B}_{\text{ID}} = \{(x_i, y_i)\}_{i=1}^{b} \sim \mathcal{D}_{\text{ID}}$ & $\mathcal{B}_{\text{OOD}} = \{x_i\}_{i=b+1}^{2b} \sim \mathcal{D}_{\text{OOD}}$
4       $\mathcal{B} = \mathcal{B}_{\text{ID}} \cup \mathcal{B}_{\text{OOD}}$
      // Feed forward the batch into $f_\theta$.
5       $[z_1, z_2, \ldots, z_{2b}] = f_\theta([x_1, x_2, \ldots, x_{2b}])$
      // Calculate the loss function with only ID representations.
6       $\mathcal{L}_{\text{LA}} = \sum_{i=1}^{b} \left[ \log[\sum_{c \in [C]} \exp(w_c^\top z_i + \log \pi_c)] - (w_{y_i}^\top z_i + \log \pi_{y_i}) \right]$
7       $\mathcal{L}_{\text{RNA}} = \sum_{i=1}^{b} [-\log(1 + \|h_\phi(z_i)\|)]$
8       $\mathcal{L} = \mathcal{L}_{\text{LA}} + \lambda \mathcal{L}_{\text{RNA}}$
      // Update the model parameters.
9       $\theta \longleftarrow \theta - \eta \nabla_\theta \mathcal{L}$
10       $W \longleftarrow W - \eta \nabla_W \mathcal{L}$
11       $\phi \longleftarrow \phi - \eta \nabla_\phi \mathcal{L}$

---

## D  DETAILS ON THE EXPERIMENTAL SETUP

### D.1  IMPLEMENTATION DETAILS

We train ResNet18 (He et al., 2015) on CIFAR10/100-LT datasets for 200 epochs with a batch size of 256. We optimize the model parameters with Adam optimizer (Kingma & Ba, 2014) with an initial learning rate of 0.001 and we decay the learning rate with the cosine learning scheduler (Loshchilov & Hutter, 2017). The weight decay parameter is set to 0.0005.

---

**Algorithm 2:** RN: Representation Norm Score

---

**Input :** Test set $\mathcal{D}_{\text{test}}$, representation extractor $f_\theta$, linear classifier $W = [w_1, w_2, \ldots, w_C]$

1   **for** $x_{\text{test}} \in \mathcal{D}_{\text{test}}$ **do**
     // Feed forward a test sample into $f_\theta$.
2      $z_{\text{test}} = f_\theta(x_{\text{test}})$
     // Calculate the RN score for OOD detection.
3      $S_{\text{RN}}(x_{\text{test}}) = -\|z_{\text{test}}\|_2$
     // Obtain the softmax prediction for ID classification.
4      $p_{\text{test}} = \arg\max \mathcal{S}(W^\top z_{\text{test}})$

---

We train ResNet50 (He et al., 2015) on ImageNet-LT for 100 epochs with a batch size of 256. We optimize the model parameters with SGD optimizer with a momentum value of 0.9 and an initial learning rate of 0.1, and we decay the learning rate with the cosine learning scheduler. The weight decay parameter is set to 0.0005.

We set the balancing hyperparameter $\lambda = 0.5$ as the default value. We do not use the learnable affine parameters $\beta$ and $\gamma$, since they are trained to amplify both ID and OOD data by converging to large values. During training, the batch consists of an equal number of ID and OOD data samples. For example, a batch size of 256 means that it contains 256 ID samples and 256 OOD samples.

For baseline methods, we implement the methods following the training setting reported in the original papers.

### D.2   COMPUTATIONAL RESOURCE AND TIME

We run the experiments on NVIDIA A6000 GPUs. For CIFAR10/100-LT datasets, we use a single GPU for each experimental run, and the entire training process takes about an hour. For ImageNet-LT, we use 7 GPUs for each experimental run, and the entire training process takes about 4 and a half hours.

### D.3   DATASETS

For the CIFAR benchmark, we employ 300K random images as an auxiliary OOD training set following Hendrycks et al. (2019a). 300K random images is a subset of 80M Tiny Images dataset (Torralba et al., 2008). The selection process for the 300K random images ensures that the image classes are disjoint with those in the CIFAR10 and CIFAR100 datasets.

For the experiments on ImageNet-1k, we use ImageNet-Extra (Wang et al., 2022b) as an auxiliary OOD training set, and ImageNet-1k-OOD, published in Wang et al. (2022b), as our OOD test set. ImageNet-Extra is created by sampling 517,711 data points from 500 randomly chosen classes in ImageNet-22k, where the classes are disjoint from the classes in ImageNet-1k. ImageNet-1k-OOD contains 50,000 OOD test images evenly sampled from 1,000 randomly selected classes of ImageNet-22k, where the 1,000 classes do not overlap with those of ImageNet-1k and ImageNet-Extra.

### D.4   BASELINE METHODS

In this section, we summarize the main baseline methods that we compare to our method in the experiments.

- **Outlier Exposure (OE) (Hendrycks et al., 2019a):** OE is a training method specifically designed for out-of-distribution (OOD) detection, with the objective of amplifying the discrepancy in softmax confidence between in-distribution (ID) and OOD data. It leverages an auxiliary OOD dataset effectively during the training process to enhance the model's ability to distinguish between ID and OOD samples. The OE method involves constructing training batches that consist of both ID training samples and auxiliary OOD samples. By combining these samples, the method aims to minimize a composite loss function that includes both the classification loss for ID samples and an additional loss term for OOD

samples. This additional loss term is defined as the cross-entropy between the uniform distribution and the softmax probabilities of the OOD samples.

- **Energy OE (Liu et al., 2020):** EnergyOE (Liu et al., 2020) is a variant of the OE method that offers an alternative approach to controlling the confidence levels of OOD samples. Instead of regularizing the cross-entropy loss of auxiliary OOD samples, EnergyOE maximizes the free energy of OOD samples. By maximizing the free energy, EnergyOE aims to increase the uncertainty and reduce the confidence associated with OOD samples, facilitating their detection and differentiation from in-distribution samples.

- **Outlier Exposure with Confidence Control (OECC) (Papadopoulos et al., 2021):** OECC is another variant of the OE method, which controls the total variation distance between the network output for OOD training samples and the uniform distribution, regularized by Euclidean distance between the training accuracy and the average confidence on the model's prediction on the training set.

- **Partial and Asymmetric Supervised Contrastive Learning (PASCL) (Wang et al., 2022b):** PASCL (Wang et al., 2022b), which is a recently developed method for OOD detection in long-tailed recognition, uses the ideas from supervised contrastive learning (Khosla et al., 2020), combined with Logit Adjustment (LA) (Menon et al., 2021) and outlier exposure (OE) (Hendrycks et al., 2019a), to achieve both high classification accuracy and reliable OOD detection performance. In particular, this method applies the partial and asymmetric contrastive learning that pushes the representations of tail-class in-distribution samples and those of OOD training samples, while pulling only tail-class in-distribution samples of the same class. In addition, in the second stage, the Batch Normalization (BN) (Ioffe & Szegedy, 2015) layers and the last linear layer are fine-tuned using only ID training data in order for the running mean and standard deviation of BN layers to be re-fit to the ID data distribution only, since the BN layers are fitted to the union of ID and auxiliary OOD data distribution before the second stage.

# E ADDITIONAL EXPERIMENTAL RESULTS

## E.1 OOD DETECTION PERFORMANCE ON SIX DIFFERENT OOD TEST SETS

Table 6: AUROC (%) for six different OOD test sets on models trained with CIFAR10/100-LT.

| Method | Texture | SVHN | CIFAR | Tiny ImageNet | LSUN | Places365 | Average |
|---|---|---|---|---|---|---|---|
| | | | ID Dataset: CIFAR10-LT | | | | |
| MSP | 72.91±1.64 | 71.98±2.82 | 70.94±0.58 | 72.86±0.66 | 74.61±0.89 | 72.80±0.72 | 72.68±1.04 |
| OECC | 91.26±0.59 | 95.86±1.00 | 80.29±0.24 | 83.69±0.09 | 90.38±0.45 | 87.02±0.21 | 88.08±0.19 |
| EnergyOE | 91.73±0.56 | 92.39±2.01 | 81.70±0.51 | 84.57±0.53 | 92.26±0.26 | 90.02±0.17 | 88.78±0.48 |
| OE | 91.66±0.72 | 95.02±1.13 | 83.43±0.32 | 86.10±0.33 | 91.81±0.62 | 90.15±0.44 | 89.69±0.51 |
| PASCL | 92.33±0.48 | 96.00±1.14 | 84.07±0.23 | 86.84±0.14 | 93.04±0.35 | 91.18±0.28 | 90.58±0.29 |
| RNA (ours) | **96.18±0.14** | **97.37±0.50** | **85.37±0.21** | **89.38±0.13** | **95.54±0.24** | **93.85±0.17** | **92.95±0.13** |
| | | | ID Dataset: CIFAR100-LT | | | | |
| MSP | 55.68±0.47 | 63.45±2.32 | 60.42±0.29 | 62.60±0.28 | 62.48±0.34 | 63.70±0.18 | 61.39±0.40 |
| OECC | 69.74±1.32 | 70.41±1.26 | 60.21±0.51 | 67.59±0.32 | 75.79±0.61 | 74.11±0.40 | 69.64±0.47 |
| EnergyOE | 69.96±0.91 | 73.46±3.93 | 61.25±0.43 | 66.29±0.23 | 72.21±0.46 | 71.66±0.20 | 69.14±0.53 |
| OE | 76.56±0.79 | 79.31±1.72 | **62.44±0.58** | **68.51±0.34** | 77.41±0.34 | 75.99±0.34 | 73.37±0.22 |
| PASCL | 76.15±0.60 | 79.23±1.39 | 62.26±0.17 | 68.37±0.25 | 77.02±0.31 | 75.79±0.24 | 73.14±0.37 |
| RNA (ours) | **79.34±0.84** | **83.01±2.71** | 57.30±0.32 | 67.96±0.29 | **80.23±0.34** | **78.17±0.23** | **74.33±0.53** |

Table 6, 7 and 8 present the AUROC, AUPR and FPR95 metrics, respectively, for the models trained using RNA and other baseline approaches across the six OOD test sets. These results collectively demonstrate the OOD detection performance. Specifically, in Table 6, RNA achieves the best results for all the six OOD test sets when the model is trained on CIFAR10-LT. For CIFAR100-LT, RNA achieves the best results on four OOD test sets as well as on average, except for CIFAR10 and Tiny ImageNet, which are semantically more similar to the training set (CIFAR100-LT) than the other OOD test sets. In Table 7, our proposed method exhibits the highest AUPR values across all the OOD

Table 7: AUPR (%) for six different OOD test sets on models trained with CIFAR10/100-LT.

| Method | OOD test set | | | | | | Average |
| | Texture | SVHN | CIFAR | Tiny ImageNet | LSUN | Places365 | |
| --- | --- | --- | --- | --- | --- | --- | --- |
| ID Dataset: CIFAR10-LT | | | | | | | |
| MSP | 54.24±1.84 | 82.52±1.56 | 66.11±0.65 | 62.76±0.71 | 70.20±0.96 | 87.05±0.42 | 70.48±0.89 |
| OECC | 87.56±0.75 | 98.28±0.45 | 80.11±0.22 | 79.83±0.14 | 90.91±0.46 | 95.22±0.09 | 88.65±0.09 |
| EnergyOE | 86.71±1.11 | 95.75±1.19 | 81.73±0.62 | 80.85±0.72 | 92.22±0.34 | 96.22±0.12 | 88.91±0.61 |
| OE | 79.52±3.63 | 96.74±1.26 | 80.18±1.04 | 78.15±0.99 | 89.03±1.70 | 95.23±0.54 | 86.47±1.41 |
| PASCL | 82.81±1.19 | 97.53±1.01 | 82.17±0.45 | 80.69±0.35 | 91.40±0.88 | 96.06±0.27 | 88.44±0.53 |
| RNA (ours) | **91.98±0.38** | **98.52±0.24** | **84.90±0.32** | **84.96±0.33** | **94.39±0.33** | **97.33±0.10** | **92.01±0.21** |
| ID Dataset: CIFAR100-LT | | | | | | | |
| MSP | 38.86±0.31 | 77.80±1.56 | **57.82±0.33** | 46.50±0.26 | 46.42±0.54 | 79.78±0.13 | 57.86±0.30 |
| OECC | 57.26±1.59 | 82.67±0.99 | 56.15±0.34 | 52.76±0.39 | 63.62±0.86 | 86.82±0.23 | 66.55±0.40 |
| EnergyOE | 58.34±1.38 | 86.22±2.22 | 57.73±0.55 | 52.02±0.42 | 59.85±0.65 | 85.89±0.16 | 66.67±0.21 |
| OE | 58.70±2.18 | 87.54±0.96 | 57.33±0.53 | 52.26±0.59 | 61.25±0.59 | 86.53±0.26 | 67.26±0.41 |
| PASCL | 57.50±1.85 | 87.09±0.75 | 57.02±0.15 | 51.85±0.44 | 60.88±0.65 | 86.31±0.18 | 66.77±0.50 |
| RNA (ours) | **68.55±1.92** | **90.35±1.67** | 53.52±0.27 | **53.37±0.43** | **67.15±0.50** | **88.57±0.21** | **70.25±0.49** |

Table 8: FPR95 (%) for six different OOD test sets on models trained with CIFAR10/100-LT.

| Method | OOD test set | | | | | | Average |
| | Texture | SVHN | CIFAR | Tiny ImageNet | LSUN | Places365 | |
| --- | --- | --- | --- | --- | --- | --- | --- |
| ID Dataset: CIFAR10-LT | | | | | | | |
| MSP | 67.27±3.88 | 61.49±5.34 | 70.52±1.56 | 65.70±0.97 | 62.11±1.33 | 66.93±1.13 | 65.67±1.82 |
| OECC | 43.70±1.45 | 24.62±3.89 | 63.02±0.79 | 56.39±0.71 | 47.77±1.26 | 55.36±0.63 | 48.48±0.88 |
| EnergyOE | 37.89±1.19 | 29.84±3.62 | 63.42±1.81 | 55.82±1.41 | 36.37±1.40 | 43.54±0.60 | 44.48±0.78 |
| OE | 23.57±0.67 | 15.74±2.37 | **56.30±0.46** | 46.26±0.93 | 27.81±0.60 | 33.19±0.59 | 33.81±0.51 |
| PASCL | 24.40±1.99 | 14.22±3.70 | 57.35±0.43 | 47.21±1.18 | 26.69±0.69 | 32.90±0.41 | 33.79±0.90 |
| RNA (ours) | **16.54±0.53** | **10.07±2.76** | 57.60±0.80 | **41.99±0.48** | **19.47±0.81** | **26.86±0.53** | **28.76±0.53** |
| ID Dataset: CIFAR100-LT | | | | | | | |
| MSP | 89.44±0.88 | 75.22±2.55 | 86.00±0.53 | 81.59±0.23 | 80.85±0.47 | 79.93±0.32 | 82.17±0.38 |
| OECC | 80.37±2.18 | 70.63±2.75 | 85.32±0.57 | 78.42±0.41 | 72.41±0.59 | 73.28±0.24 | 76.74±0.58 |
| EnergyOE | 84.40±1.04 | 75.12±5.22 | 81.45±0.48 | 81.15±0.64 | 79.64±1.20 | 78.50±0.76 | 80.04±1.13 |
| OE | 67.08±2.13 | 55.36±3.62 | 79.85±0.71 | **76.17±0.83** | 63.24±0.31 | 65.31±0.40 | 67.83±0.82 |
| PASCL | **65.50±1.64** | 53.57±3.21 | **79.81±0.35** | 76.37±0.38 | 64.09±0.88 | 64.83±0.41 | **67.36±0.46** |
| RNA (ours) | 68.67±1.66 | **51.48±6.51** | 84.92±0.36 | 79.90±0.41 | **60.53±0.59** | **64.06±0.17** | 68.26±0.94 |

test sets when trained on CIFAR10-LT. Similarly, for all but the CIFAR10 test set, RNA achieves the highest AUPR values when trained on CIFAR100-LT. In Table 8 our proposed method shows the best FPR95 values across all but the CIFAR100 test sets when trained on CIFAR10-LT. On the other hand, while RNA achieves the lowest FPR95 values for SVHN, LSUN, and Places365 when trained on CIFAR100-LT, it is outperformed by PASCL in terms of the average FPR95 value across the OOD test sets.

Near OOD detection poses a significant challenge in the field of OOD detection. In these scenarios, the OOD test sets closely resemble the ID datasets. RNA method exhibits suboptimal performance for near OOD detection as shown in Table 6, 7 and 8. This is particularly noticeable when using CIFAR100-LT as the ID training set and CIFAR10 as the OOD test set. In contrast, other methods like OE with MSP score exhibit relatively robust performance in such settings.

We attribute this observed trend to the inherent characteristics of the CIFAR10 and CIFAR100 datasets. While these datasets consist of disjoint classes, certain classes share overlapping features. For instance, presenting data from the "dog" class of CIFAR10 OOD test set to a CIFAR100-trained model may activate features associated with CIFAR100 classes like "fox", "raccoon", "skunk", "otter", and "beaver". Consequently, the logits corresponding to these multiple classes may simultaneously have high values, potentially leading to a low MSP value. However, the representation norm might be large due to these related features. For such an example, the near OOD sample may be detected by MSP but incorrectly categorized as an ID sample by the RN score. To support this hypothesis, Table

Table 9: The AUROC performance (%) of the model trained with RNA using different OOD scoring method. The OOD test sets for near OOD task are CIFAR10 and Tiny ImageNet, while those for far OOD task are Texture, SVHN, LSUN, and Places365.

| Training method | Scoring method | Near OOD | Far OOD | Average |
|---|---|---|---|---|
| OE | MSP | **65.48** | 77.32 | 73.37 |
| RNA | MSP | 64.81 | 78.49 | 73.93 |
| RNA | RN | 62.63 | **80.19** | **74.33** |

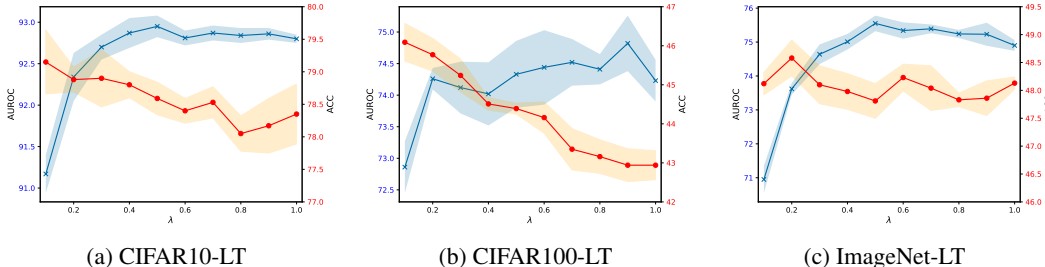

|  (a) CIFAR10-LT | (b) CIFAR100-LT | (c) ImageNet-LT |
|---|---|---|

Figure 3: AUROC and ACC (%) performance of RNA-trained model with varying the balancing hyperparameter $\lambda$. Blue (x-marked) lines indicate the AUROC values and red (circle-marked) lines indicate the ACC values.

9 illustrates the OOD detection performance (AUROC) for two OOD scoring methods, MSP and RN, when applied to the RNA-trained model on CIFAR100-LT. As anticipated, using MSP leads to an improvement in near OOD detection performance, but at the cost of worsening far OOD detection.

## E.2  HYPERPARAMETER ROBUSTNESS

Our proposed training method uses the combination of the two loss terms, the classification loss and the RNA loss regularizing the representation norm of input data, as shown in equation Eq. 4. The relative importance of the two loss terms can be tuned with the hyperparameter $\lambda$. Although all the results in the main paper are obtained with the fixed value of $\lambda = 0.5$, we extensively explore the performance results when models are trained with varying the hyperparameter $\lambda$ for CIFAR10-LT, CIFAR100-LT, and ImageNet-LT. In Figure 3, we plot the AUROC and ACC performances for 10 values of $\lambda = 0.1, 0.2, \ldots, 1.0$. For CIFAR10/100-LT and ImageNet-LT, the AUROC values (blue lines) are robust to the change in $\lambda$ when $\lambda$ is greater than $0.2$, although they are much lower when $\lambda$ is small. On the other hand, the classification accuracy (red lines) decreases as $\lambda$ increases when trained on CIFAR10-LT and CIFAR100-LT, and this tendency is more dramatic when trained on CIFAR100-LT. For ImageNet-LT, the accuracy does not change significantly as $\lambda$ values change.

## E.3  IMBALANCE RATIO

Table 10: AUROC (%) and classification accuracy (%) of models trained on CIFAR10-LT and CIFAR100-LT with various imbalance ratio $\rho$.

| Method | CIFAR10-LT | | | | | CIFAR100-LT | | | |
|---|---|---|---|---|---|---|---|---|---|
| | AUROC / ACC | | | | | | | | |
| $\rho$ | 1 | 10 | 50 | 100 | 1000 | 1 | 10 | 50 | 100 |
| OE | 96.78/93.55 | 94.32/87.47 | 91.35/79.62 | 89.69/74.85 | 82.98/57.21 | 84.56/71.32 | 78.57/55.78 | 74.72/44.13 | 73.37/39.83 |
| PASCL | 96.67/93.67 | 94.68/87.62 | 92.25/80.15 | 90.58/76.39 | 86.62/59.43 | **84.64**/71.89 | **78.59**/57.89 | 75.02/47.16 | 73.14/43.44 |
| RNA (ours) | **97.07/94.24** | **95.86/89.28** | **93.87/82.29** | **92.95/78.59** | **88.93/60.72** | 81.79/**74.10** | 78.19/**60.17** | **75.67/48.81** | **74.33/44.39** |

In Table 10, we present the AUROC and classification accuracy (ACC) metrics for the models trained on CIFAR10/100-LT with different imbalance ratios. The maximum number of training samples per class is set to 5,000 and 500 for CIFAR10 and CIFAR100, respectively. Consequently, we

can construct CIFAR10-LT with an imbalance ratio 1,000, where the minimum number of training samples per class is 5. However, CIFAR100-LT with an imbalance ratio 1,000 cannot be constructed due to the number of samples. Since the imbalance ratio indicates how imbalanced the training set is, it is more challenging to train a model with a higher imbalance ratio. It is worth noting that an imbalance ratio of 1 indicates a balanced training dataset rather than a long-tailed dataset.

In particular, when trained on CIFAR10-LT, our proposed method consistently outperforms PASCL in terms of both AUROC and ACC across the imbalance ratios of 1, 10, 50, 100, and 1,000, including the task with a balanced training set. On the other hand, when trained on CIFAR100-LT, RNA improves the accuracy of the model across all the imbalance ratios, but it exhibits relatively lower AUROC when the imbalance ratio is 1 or 10. These results highlight that our proposed model is superior for various imbalance ratios of the training set, especially for high imbalance ratios.

### E.4    ABLATION STUDY ON THE VARIANTS OF RNA TRAINING LOSS

Table 11: OOD detection and ID classification performance (%) on CIFAR10-LT with variants of RNA loss. The gray regions indicate the results for our original proposed method (RNA).

| Training loss | | | AUROC(↑) | AUPR(↑) | FPR95(↓) | ACC(↑) | Many(↑) | Medium(↑) | Few(↑) |
|---|---|---|---|---|---|---|---|---|---|
| $\mathcal{L}_{LA}$ | $\mathcal{L}_{OOD\text{-attenuation}}$ | $\mathcal{L}_{ID\text{-amplification}}$ | | | | | | | |
| ✓ | ✓ | | 82.68 | 90.99 | 55.05 | 77.82 | **94.12** | 74.63 | 65.49 |
| ✓ | ✓ | ✓ | 84.03 | 76.62 | 41.59 | 72.92 | 82.26 | 68.55 | **69.62** |
| ✓ | | ✓ | **92.95** | **92.01** | **28.76** | **78.59** | 93.43 | **74.92** | 68.25 |

We conduct an ablation study to asses the effectiveness of different variants of the RNA loss. Our proposed RNA loss ($\mathcal{L}_{ID\text{-amplification}}$) involves amplifying ID representation norms during training. We examine two alternative variants: one involves a loss to attenuate the norms of OOD samples, referred to as $\mathcal{L}_{OOD\text{-attenuation}}$, and the other is a combination of both losses. The OOD-attenuation loss is defined as $\mathcal{L}_{OOD\text{-attenuation}} = \mathbb{E}_{x \sim \mathcal{D}_{OOD}} [\log(1 + \|h_\phi(f_\theta(x))\|)]$, which is the negative counterpart of the original RNA loss in Equation 3. As shown in Table 11, the two variants turn out to be not as effective as the original RNA loss, both in OOD detection and ID classification for CIFAR10-LT training dataset.

Table 12: Representation norm and the activation ratio at the last ReLU layer with variants of RNA loss on ID test set (CIFAR10) AND OOD test set (SVHN). The gray regions indicate the results for our original proposed method (RNA).

| Training loss | | | Representation norm | | Activation ratio | |
|---|---|---|---|---|---|---|
| $\mathcal{L}_{LA}$ | $\mathcal{L}_{OOD\text{-attenuation}}$ | $\mathcal{L}_{ID\text{-amplification}}$ | ID test | OOD test | ID test | OOD test |
| ✓ | ✓ | | 28.84 | 14.55 | 0.55 | 0.55 |
| ✓ | ✓ | ✓ | 43.65 | 13.14 | 0.74 | 0.18 |
| ✓ | | ✓ | 45.90 | 0.93 | 0.66 | 0.02 |

As shown in Table 12 the combination of attenuation loss and amplification loss widens the gap in representation norm and activation ratio compared to the case of using only the attenuation loss. However, employing the amplification loss alone creates a more substantial gap than the combination loss does. In particular, we can see that the combination loss succeeds in enlarging the ID norms, but whenever attenuation loss is used (either alone or with amplification loss), the resulting OOD norm is not as low as the case of using only the amplification loss.

The rationale behind this result lies in the optimization process. When solely employing the amplification loss (RNA loss, defined as Equation 3), the optimizer amplifies the norm $\|h_\phi(f_\theta(x))\|$ for ID data, necessarily by promoting high activation ratios for ID data while concurrently reducing the activation ratio for OOD data (to be as low as 2% in our experiment), due to the BN regularization using both ID and OOD data.

On the other hand, when the attenuation loss $\mathbb{E}_{x \sim \mathcal{D}_{OOD}} [\log(1 + \|h_\phi(f_\theta(x))\|)]$ is applied (either alone or in combination form), the optimizer does not necessarily suppress activations for OOD samples in the feature coordinates to minimize the loss. Instead, the loss can be simply minimized by finding that maps the OOD features into its null space. Due to this additional possibility, the resulting

Table 13: OOD detection and ID classification performance (%) on CIFAR10-LT with ImageNet-RC (Chrabaszcz et al., 2017) as the auxiliary OOD training set.

| Method | AUROC ($\uparrow$) | AUPR ($\uparrow$) | FPR95 ($\downarrow$) | ACC ($\uparrow$) | Many ($\uparrow$) | Medium ($\uparrow$) | Few ($\uparrow$) |
|---|---|---|---|---|---|---|---|
| OE | 92.29 | **91.85** | 28.16 | 74.67 | **94.59** | 73.64 | 56.10 |
| PASCL | 92.05 | 91.70 | 29.72 | 73.58 | 94.43 | 72.76 | 53.88 |
| PASCL (fine-tune) | 90.66 | 89.44 | 33.88 | 77.78 | 93.53 | 73.88 | 67.12 |
| RNA (ours) | **92.53** | 91.78 | **23.61** | **78.67** | 94.13 | **75.44** | **67.67** |

activation ratios and the representation norms for OOD samples are not as low as the case of using only the amplification loss.

In summary, our analysis and simulation results show that directly applying the attenuation loss may not necessarily create a significant gap in either activation ratio or representation norm between ID and OOD data. While utilizing a combined loss yields better results than using only the attenuation loss, it still falls short of the performance achieved by the RNA loss. This is because the combined loss does not inherently lead to a solution where high activation for ID and low activation for OOD data are simultaneously achieved.

### E.5   ALTERNATIVE AUXILIARY OOD TRAINING SET

We additionally conduct experiments employing ImageNet-RC as an auxiliary OOD dataset for CIFAR10-LT ID dataset. The results are presented in Table 13. Our results demonstrate the effectiveness of the RNA, which consistently outperforms both the OE and PASCL, in terms of OOD detection and ID classification. While the AUROC and AUPR metrics exhibited comparable performance across the three methods, the FPR95 metric indicated RNA's superiority, surpassing OE and PASCL by 4.55% and 6.11%, respectively. Furthermore, RNA demonstrats a 4.00% enhancement in ID classification accuracy compared to OE, and a 0.89% improvement over PASCL. These results validate RNA's effectiveness not only when trained with the 300K random images but also when employed with other OOD auxiliary sets.

### E.6   STRUCTURE OF PROJECTION FUNCTION

Table 14: OOD detection and ID classification performance (%) on CIFAR10-LT with various structure of projection function $h_\phi$. The gray regions indicate the results for our original proposed method (RNA).

| Training objective | AUROC($\uparrow$) | AUPR($\uparrow$) | FPR95($\downarrow$) | ACC($\uparrow$) | Many($\uparrow$) | Medium($\uparrow$) | Few($\uparrow$) |
|---|---|---|---|---|---|---|---|
| $\|f_\theta(x)\|$ | 92.18 | 92.27 | 33.54 | 80.11 | 94.07 | 77.25 | 69.87 |
| $\|W_h^\top f_\theta(x)\|$ | 92.78 | 91.89 | 29.48 | 77.63 | 93.67 | 74.02 | 66.10 |
| $\|h_\phi(f_\theta(x))\|$ | 92.95 | 92.01 | 28.76 | 78.59 | 93.43 | 74.92 | 68.25 |

Table 15: The average norm of representations of ID training set (CIFAR10-LT), OOD training set (300K random images), ID test set (CIFAR10), and OOD test set (SVHN) with various structure of projection function $h_\phi$. The gray regions indicate the results for our original proposed method (RNA).

| Training objective | Norm of | ID train | OOD train | ID test | OOD test |
|---|---|---|---|---|---|
| $\|f_\theta(x)\|$ | | 79.89 | 7.93 | 59.31 | 4.61 |
| $\|W_h^\top f_\theta(x)\|$ | $\|f_\theta(x)\|$ | 66.29 | 2.59 | 46.32 | 1.30 |
| $\|h_\phi(f_\theta(x))\|$ | | 65.14 | 2.53 | 45.90 | 0.93 |

We conduct an ablation study to evaluate the impact of architectural variations, specifically focusing on the structure of the projection function $h_\phi$. In the proposed approach, we employ a 2-layer MLP configuration for $h_\phi$. As alternative configurations, we train the model using RNA loss on CIFAR10-LT, where in Equation 3, $\|h_\phi(f_\theta(x))\|$ is replaced either by the feature norm $\|f_\theta(x)\|$ or

the linearly projected norm $\|W_h^\top f_\theta(x)\|$. The results presented in Table 14 indicate that the structure of the projection function $h_\phi$ is not a critical element in our method, as the OOD detection and ID classification performances remain comparable across the model variants. The reason we use the 2-layer MLP projection function in our proposed approach is to regulate the situations where the feature norm of ID samples increases excessively during training. Nevertheless, our findings indicate that employing the feature norm itself within the logarithmic term adequately prevents the divergence of $\|f_\theta(x)\|$, as shown in the Table 15.

### E.7 FINE-TUNING TASK WITH AUXILIARY OOD DATA

In the fine-tuning task for OOD detection (Hendrycks et al., 2019a; Liu et al., 2020; Papadopoulos et al., 2021), a classification model is initially trained without the OOD set and then later fine-tuned with the OOD set to improve its OOD detection ability.

We evaluate the effectiveness of our training method when applied to the fine-tuning task. The pre-trained models are obtained by training a model on CIFAR10/100-LT and ImageNet-LT, respectively, using only the classification loss, either CE or LA. We then train this model for 10 (respectively 3) more epochs on CIFAR10/100-LT and 300K random images (respectively ImageNet-LT and ImageNet-Extra) with an objective of $\mathcal{L}_{\text{ID}} + \lambda\mathcal{L}_{\text{OOD}}$. Here, we use $\mathcal{L}_{\text{CE}}$ or $\mathcal{L}_{\text{LA}}$ as $\mathcal{L}_{\text{ID}}$, and $\mathcal{L}_{\text{OE}}$, $\mathcal{L}_{\text{EnergyOE}}$, or $\mathcal{L}_{\text{RNA}}$ as $\mathcal{L}_{\text{OOD}}$. Table 16 summarizes the results. "PT" and "FT" stand for pre-training and fine-tuning methods, respectively. We can oberve that RNA attains the most optimal performances in both OOD detection and ID classification. Notably, when the classification loss during pre-training or fine-tuning is LA, the classification accuracy of models fine-tuned using OE or EnergyOE is significantly lower than that of RNA. This discrepancy could be attributed to the inherent trade-offs between OOD detection and ID classification, arising from the application of OE as discussed in Sec. 3.1.

### E.8 TRAINING DYNAMICS OF RNA

Figure 4a and 4b depict the changes of training losses (LA loss $\mathcal{L}_{LA}$ and RNA loss $\mathcal{L}_{RNA}$), respectively, over the training. Figure 4c and 4d illustrate the dynamics of representation norms of training ID, test ID, training OOD, and test OOD data, over the training. These figures demonstrate that the training effectively works to widen the gap between ID and OOD representation norms.

### E.9 REPRESENTATION NORM DSITRIBUTION

We present the histograms of representation norms for models trained with LA and RNA, respectively, to visualize the impact of RNA loss on the representation norm distribution. Figure 5 displays the histograms of representation norms of ID test data and OOD test data on LA/RNA-trained models with CIFAR10-LT and CIFAR100-LT datasets. The evident gap in representation norms between ID test set and OOD test set is shown in the figure for RNA-trained models. This gap in the distributions of representation norms of ID and OOD data enables effective OOD detection using representation norms.

Table 16: The results of fine-tuning with auxiliary OOD data on pre-trained models by CIFAR10-LT, CIFAR100-LT, and ImageNet-LT. "PT"/"FT" refer to pre-training and fine-tuning, resp., where the fine-tuning loss is $\mathcal{L}_{\text{ID}} + \lambda\mathcal{L}_{\text{OOD}}$.

| PT CE | PT LA | FT ($\mathcal{L}_{\text{ID}}$) CE | FT ($\mathcal{L}_{\text{ID}}$) LA | FT ($\mathcal{L}_{\text{OOD}}$) | AUROC (↑) | AUPR (↑) | FPR95 (↓) | ACC (↑) |
|---|---|---|---|---|---|---|---|---|
| | | | | | ID dataset: CIFAR10-LT | | | |
| ✓ | | ✓ | | | 90.23 | 90.15 | 39.51 | 76.82 |
| ✓ | | | ✓ | OE | 90.36 | 90.05 | 36.70 | 76.26 |
| | ✓ | | ✓ | | 90.29 | 89.89 | **36.69** | 76.21 |
| ✓ | | ✓ | | | 88.78 | 88.91 | 44.48 | 76.25 |
| ✓ | | | ✓ | EnergyOE | 85.10 | 83.36 | 49.01 | 70.28 |
| | ✓ | | ✓ | | 87.08 | 86.73 | 48.97 | 71.45 |
| ✓ | | ✓ | | | 90.34 | 90.56 | 39.87 | 74.28 |
| ✓ | | | ✓ | RNA (ours) | 90.52 | 90.74 | 39.39 | **78.78** |
| | ✓ | | ✓ | | **90.53** | **90.85** | 36.70 | 78.53 |
| | | | | | ID dataset: CIFAR100-LT | | | |
| ✓ | | ✓ | | | 71.26 | 66.82 | 72.58 | 40.27 |
| ✓ | | | ✓ | OE | 70.91 | 66.52 | 72.57 | 41.09 |
| | ✓ | | ✓ | | 71.82 | 67.27 | 71.08 | 41.06 |
| ✓ | | ✓ | | | 69.18 | 66.24 | 77.23 | 39.46 |
| ✓ | | | ✓ | EnergyOE | 68.73 | 65.93 | 78.14 | 38.32 |
| | ✓ | | ✓ | | 69.13 | 66.35 | 77.61 | 38.24 |
| ✓ | | ✓ | | | **73.02** | **68.93** | **70.65** | 40.69 |
| ✓ | | | ✓ | RNA (ours) | 72.58 | 68.72 | 71.99 | **44.82** |
| | ✓ | | ✓ | | 72.56 | 68.85 | 71.67 | 44.65 |
| | | | | | ID dataset: ImageNet-LT | | | |
| ✓ | | ✓ | | | 63.55 | 65.34 | 89.63 | 32.28 |
| ✓ | | | ✓ | OE | 64.68 | 65.66 | 88.05 | 37.23 |
| | ✓ | | ✓ | | 63.80 | 65.45 | 89.17 | 35.78 |
| ✓ | | ✓ | | | 63.81 | 65.13 | 88.29 | 38.43 |
| ✓ | | | ✓ | EnergyOE | 47.63 | 46.77 | 92.82 | 31.79 |
| | ✓ | | ✓ | | 36.47 | 40.31 | 95.59 | 19.70 |
| ✓ | | ✓ | | | 68.95 | 68.94 | 86.05 | 34.66 |
| ✓ | | | ✓ | RNA (ours) | 69.27 | 69.22 | 85.37 | **38.91** |
| | ✓ | | ✓ | | **72.20** | **71.45** | **83.25** | 37.66 |

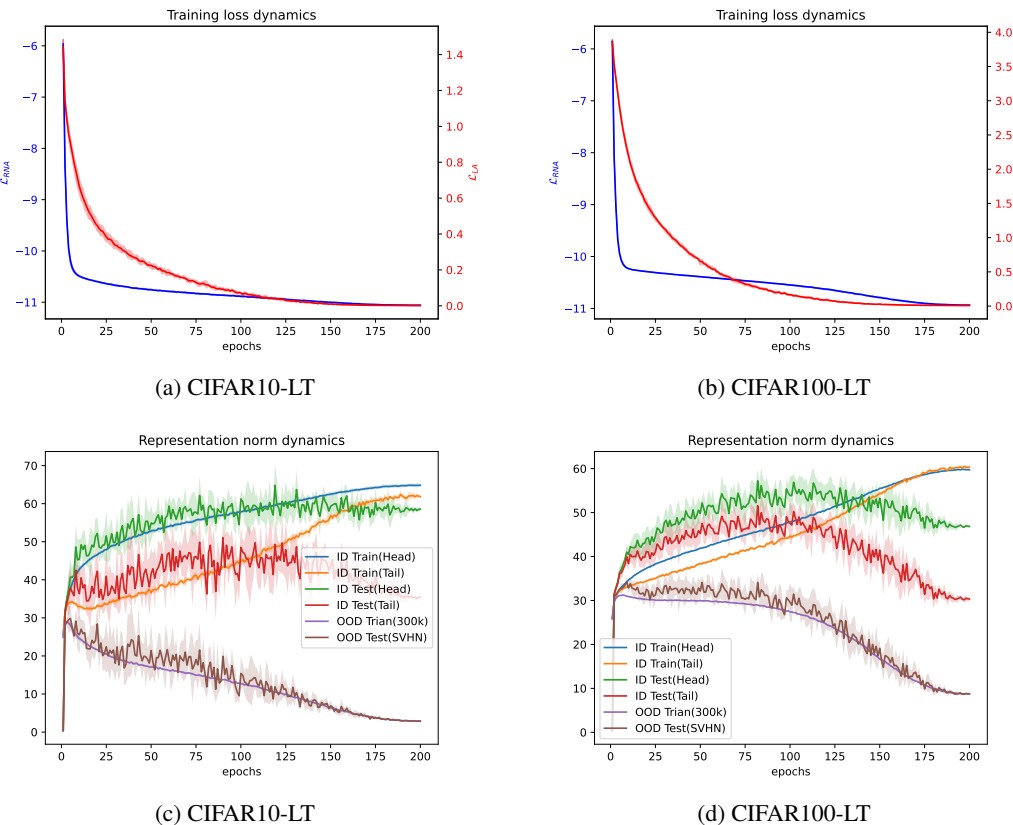

(a) CIFAR10-LT         (b) CIFAR100-LT

(c) CIFAR10-LT         (d) CIFAR100-LT

Figure 4: The training dynamics of the models trained with RNA on CIFAR10-LT and CIFAR100-LT. The training loss dynamics of LA loss (red) and RNA loss (blue) are in (a) and (b), and the representation norm dynamics of head and tail data in ID training and test dataset, OOD training set (300k random images), and OOD test set(SVHN) are presented in (c) and (d).

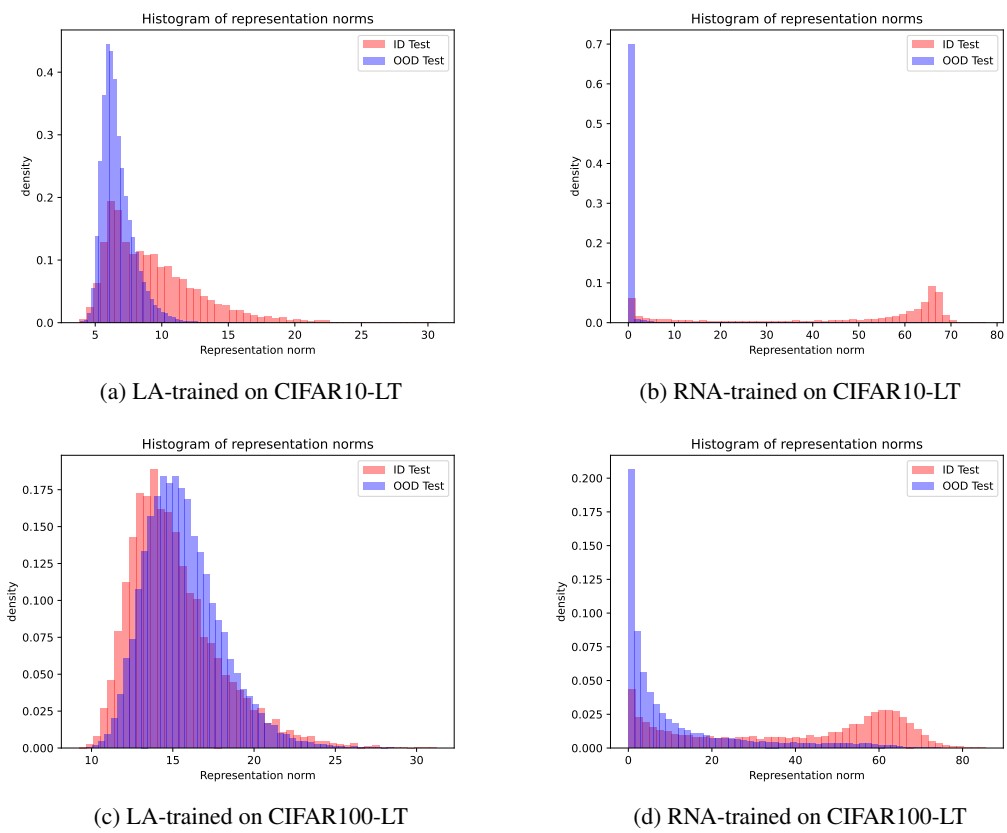

(a) LA-trained on CIFAR10-LT

(b) RNA-trained on CIFAR10-LT

(c) LA-trained on CIFAR100-LT

(d) RNA-trained on CIFAR100-LT

Figure 5: The histogram of representation norms of ID test data and OOD test data on LA/RNA-trained models with CIFAR10-LT and CIFAR100-LT. The red bars represent the density of representation norms of ID test data and the blue bars represent that of OOD test data(SVHN). The evident gap in representation norms between ID test set and OOD test set is shown in the figure for RNA-trained models. This gap in the distributions of representation norms of ID and OOD data enables effective OOD detection using representation norms.

