# OpenReview forum: "Representation Norm Amplification for Out-of-Distribution Detection in Long-Tail Learning"
_ICLR.cc/2024/Conference — ICLR 2024 Conference Withdrawn Submission_

### Official Review · Reviewer_6DDx · 2023-10-19

**Soundness:** 3 good
**Presentation:** 3 good
**Contribution:** 2 fair
**Rating:** 5
**Confidence:** 3

**Summary:**

The paper tackles the significant issue of out-of-distribution (OOD) detection within the realm of long-tail learning. It introduces an innovative approach known as Representation Norm Amplification (RNA), with the goal of separating OOD detection from classification. This effectively resolves the trade-offs between OOD detection and in-distribution (ID) classification commonly encountered by existing methods. The main idea of RNA relies on the norm of representation vectors as a new dimension to differentiate between ID and OOD data, ultimately delivering superior performance in both OOD detection and classification.

**Strengths:**

(1) This work is well-organized and features a clear presentation. The problem statement concerning the challenges of out-of-distribution (OOD) detection in the context of long-tail learning is precisely defined and effectively underscores the motivation behind the proposed method RNA.

(2) The RNA approach proposed in this paper, which leverages representation norms to separate OOD detection from classification, is both innovative and logically grounded. This approach addresses the trade-offs encountered in existing methods and exhibits enhanced performance.

(3) The study presents comprehensive experimental results across multiple datasets, showcasing the effectiveness of RNA in improving both OOD detection and classification accuracy.

**Weaknesses:**

(1)  This work lacks a more detailed description of the proposed RNA method, such as a pseudocode algorithm, which would enhance the clarity of the paper and help explain the key steps of the proposed approach.

(2) While the paper mentions existing methods, it would significantly benefit from a comparison with more recent state-of-the-art work in both OOD detection and long-tail learning, e.g., [1, 2]. This would elevate the quality and relevance of the paper, providing a more up-to-date context for the proposed method.

(3)  Including additional explanations and visualizations of how representation norms are amplified of the network architecture would improve the interpretability of the model's decisions. This would make the paper more valuable by providing insights into the inner workings of the RNA method.

(4) The work is expected to consider a more comprehensive evaluation of larger and more complex datasets, such as iNaturalist [3]. This would provide a broader assessment of RNA's performance and its applicability to real-world scenarios.

[1] Out-of-Distribution Detection with Deep Nearest Neighbors, ICML 2022

[2] POEM: Out-of-Distribution Detection with Posterior Sampling, ICML 2022

[3] The iNaturalist Species Classification and Detection Dataset, CVPR 2018

**Questions:**

(1) Could the authors provide a pseudocode algorithm or a more detailed step-by-step explanation of the RNA method? This would enhance the clarity of the practical implementation of the approach.

(2) It would be valuable if the paper could incorporate a comparison with more recent state-of-the-art methods in both OOD detection and long-tail learning to provide a more comprehensive context for the significance of the proposed RNA method.

(3) Could the authors offer additional explanations or visualizations illustrating how representation norms are amplified in the RNA method and how this amplification process influences the network's decisions? This would contribute to a better understanding of the model's interpretability.

(4) The paper discusses the trade-offs between OOD detection and long-tail recognition. It would be beneficial for the authors to provide further elaboration on how RNA specifically addresses these trade-offs and achieves a balance between the two objectives. Additionally, insights into how RNA's performance varies with different levels of class imbalance in the long-tail datasets would be valuable.

---

> ### Author Response · Authors · 2023-11-18
> **Response to 6DDx (1/2)**
>
> Thank you for providing valuable feedback. We appreciate the constructive feedback and hope our answers would clear up your concerns.
>
> **1. Pseudocode**
> >(Q1) Could the authors provide a pseudocode algorithm or a more detailed step-by-step explanation of the RNA method?
>
> The pseudocode for our proposed method is available in Appendix C.
>
> **2. Other OOD scoring methods**
> >(Q2) It would be valuable if the paper could incorporate a comparison with more recent state-of-the-art methods in both OOD detection and long-tail learning to provide a more comprehensive context for the significance of the proposed RNA method.
>
> The table below reports the OOD detection performance (AUROC) of CE, LA, and RNA-trained models employing various OOD scoring methods, including MSP, Energy, RN, and ASH[f], which is one of the SOTA OOD scoring methods.
> LA+ASH outperforms LA+Energy by 2.47% and 9.24% on CIFAR100-LT and ImageNet-LT, respectively. However, LA+ASH performs worse than RNA+ASH by 7.22% and 6.00% on CIFAR100-LT and ImageNet-LT, respectively, and even worse than RNA+MSP on CIFAR100-LT. This result shows that applying a post-hoc OOD detection method combined with a long-tail learning method may not hurt the ID classification but is not as effective as exposing auxiliary OOD in improving the OOD detection performance.
> We also notice that our RN scoring, which uses the representation norm for OOD detection, outperforms other scoring methods including ASH for RNA-trained models. This implies that the representation norm can encapsulate information useful for OOD detection, and our proposed training method effectively embeds such information about OOD-ness into the representation norm.
>
> |  | MSP | Energy | RN | ASH |
> |---|:---:|:---:|:---:|:---:|
> | ID Dataset: CIFAR100-LT |  |  |  |  |
> | CE | 61.39 | 62.65 | 55.67 | 62.78 |
> | LA | 62.86 | 62.46 | 54.20 | 64.93 |
> | RNA | 73.93 | 74.00 | 74.33 | 72.15 |
> | ID Dataset: ImageNet-LT |  |  |  |  |
> | CE | 54.23 | 52.03 | 56.73 | 59.07 |
> | LA | 55.67 | 53.30 | 54.89 | 62.54 |
> | RNA | 61.27 | 61.50 | 75.55 | 68.54 |
>
> [f] Djurisic et al., Extremely Simple Activation Shaping for Out-of-Distribution Detection, ICLR 2023.
>
> **3. Rationale behind RNA**
> >(Q3) Could the authors offer additional explanations or visualizations illustrating how representation norms are amplified in the RNA method and how this amplification process influences the network's decisions? This would contribute to a better understanding of the model's interpretability.
>
> In Appendix E.8 in the revised paper, Figure 4 (a,b) depicts the changes of training losses (LA loss $\mathcal{L}_ {\text{LA}}$ and RNA loss $\mathcal{L}_ {\text{RNA}}$) over the training, and Figure 4 (c,d) illustrates the dynamics of representation norms of training ID, test ID, training OOD, and test OOD data. These figures demonstrate that the training effectively works to widen the gap between ID and OOD representation norms.
> The rationale behind these outcomes lies in the interaction of the last BN layer, ReLU layer, and the RNA loss. Since the BN layer subtracts the estimated sample mean from input vectors, the elements of the subtracted vectors at each coordinate are  approximately half positive and half negative. Upon passing through the subsequent ReLU layer, approximately half of them are activated, and the rest are deactivated.
> During training with RNA loss, among the input vectors of the last BN layer, ID vectors are enlarged while OOD vectors are not. Thus, as the training proceeds, the following inequalities emerge:
> $f_ {\text{before BN}}(x_ {\text{OOD}})_ k < \mu_ {\text{BN}} < f_ {\text{before BN}}(x_ {\text{ID}})_ k$
> for many $k$’s, where $f_ {\text{before BN}}(x)$ represents the latent vector right before the last BN layer, and $f_ {\text{before BN}}(x)_ k$ represents the $k$-th coordinate of $f_ {\text{before BN}}$.
> As a result, passing through the last BN-ReLU layer, many coordinates are activated and enlarged for ID representations, while those are deactivated for of OOD representations. This difference of activation ratios is clearly illustrated in Figure 2(b) in the original paper. The resulting gap in activation ratios directly translates to a gap in representation norms. Consequently, we can leverage the representation norms for effective OOD detection.
>
> We also present the histograms of representation norms for models trained with LA and RNA to visualize the impact of RNA loss on the representation norm distribution. Figure 5 in Appendix E.9 displays the histograms of representation norms of ID test data and OOD test data  on LA/RNA-trained models with CIFAR10-LT and CIFAR100-LT datasets. The evident gap in representation norms between ID test set and OOD test set is shown in the figure for RNA-trained models. This gap in the distributions of representation norms of ID and OOD data enables effective OOD detection using representation norms.

---

> ### Author Response · Authors · 2023-11-18
> **Response to Reviewer 6DDx (2/2)**
>
> **4. Test with more OOD datasets**
> >(W4) The work is expected to consider a more comprehensive evaluation of larger and more complex datasets, such as iNaturalist [3]. This would provide a broader assessment of RNA's performance and its applicability to real-world scenarios.
>
> Following the reviewer's suggestion, we expanded our evaluation of OOD detection performance on additional test sets, namely iNaturalist, SUN, Places, Texture, and OpenImage-O. The table below summarizes the AUROC performance values. While our method shows superior performance on iNaturalist, SUN, and Places, it exhibits a performance gap compared to CE, OE, and PASCL models on Texture and OpenImage-O. Notably, on the Texture and OpenImage-O datasets, both OE and PASCL perform worse than the CE baseline model, and our proposed model exhibits a even lower performance. However, when we use MSP score for RNA-trained model on Texture and OpenImage-O, the AUROC values increase by 14.22% and 8.92% on Texture and OpenIamge-O, respectively, compared to using the RN score, and recovers the performance drop or even achieves the best result. Thus, we hypothesize that the primary issue lies in using RN (representation norm) score as the OOD scoring method on Texture dataset and OpenImage-O. We conjecture that this may be attributed from the characteristics of test OOD datasets, and plan to further examine why the representation norm scoring is ineffective for certain OOD test datasets by analyzing the characteristics of the datasets such as inherent overlap or similarity to ID datasets.
>
> | Method | iNaturalist | SUN | Places | Texture | OpenImage-O |
> |---|:---:|:---:|:---:|:---:|:---:|
> | CE | 74.81 | 67.32 | 66.85 | **63.65** | 64.33 |
> | OE | 72.28 | 70.25 | 70.80 | 61.30 | 62.59 |
> | PASCL | 72.63 | 71.15 | 71.57 | 62.45 | 63.25 |
> | RNA (ours) | **81.28** | **73.88** | **74.33** | 45.97 | 57.17 |
> | RNA/MSP | 76.70 | 70.35 | 70.42 | 61.81 | **66.09** |
>
> **5. RNA overcomes the trade-offs between OOD detection and LT learning**
> >(Q4) The paper discusses the trade-offs between OOD detection and long-tail recognition. It would be beneficial for the authors to provide further elaboration on how RNA specifically addresses these trade-offs and achieves a balance between the two objectives.
>
> In Section 2, we investigate the trade-offs inherent in methods designed for OOD detection and LT learning contexts. In OOD detection context, the training scheme discourages the confidence predictions on rare OOD data. Conversely, in the LT learning context, the training scheme encourages confident predictions on rare tail-class in-distribution samples. Thus, combining these methods generates the trade-offs between OOD detection and long-tail recognition. However, we propose that these trade-offs can be mitigated by introducing a new dimension, the norm of representation, for OOD detection, while controlling the confidence levels only for LT learning.
> In Table 1, when the conventional OOD detection method is applied such as CE+OE and LA+OE, the confidence values of Few (tail) classes are notably low. In contrast, the RNA-trained model exhibits high confidence values for tail classes as outperforming in OOD detection. Moreover, since the confidence is controlled solely for LT learning, overall and Few accuracy is higher.
> To summarize, trade-offs exist in prior methods tailored exclusively for either OOD detection or LT learning. Our goal is not to identify the optimal trade-off but to transcend these trade-offs, enhancing OOD detection and classification performance. Furthermore, the decoupling of OOD detection and classification demonstrates a positive impact on the calibration performance, as evidenced in Table 5.
>
>
> **6. Imbalance ratio**
> >(Q4) Additionally, insights into how RNA's performance varies with different levels of class imbalance in the long-tail datasets would be valuable.
>
> We performed experiments across various imbalance ratio values and reported the results in Appendix E.3. Please refer to Table 10. In particular, we reported the AUROC and classification accuracy (ACC) metrics for the models trained on CIFAR10/100-LT with different imbalance ratios, including the balanced case $\rho=1$. When trained on CIFAR10-LT, our proposed method consistently outperforms OE in terms of both AUROC and ACC across all the imbalance ratios of 1, 10, 50, 100, and 1,000, including the task with a balanced training set. When trained on CIFAR100-LT, RNA improves the accuracy of the model across all the imbalance ratios, but it exhibits relatively lower AUROC than OE when the imbalance ratio is as low as 1 or 10.

---

> > ### Comment · Reviewer_6DDx · 2023-11-23
> >
> > Thank you for the detailed response. I have reviewed the response and the discussions with other reviewers.
> >
> > I appreciate the explanation regarding RNA and the discussion of the trade-offs inherent in methods designed for OOD detection and LT learning. I think the paper has been improved during the rebuttal process. However, I share some similar concerns with reviewer MX6w that the idea seems somewhat general for OOD detection. This underscores the importance of comparing it with state-of-the-art works in OOD detection, such as KNN[1] and POEM[2], which have not been addressed in the response.
> >
> > Overall, I choose to maintain my current evaluation score.

---

### Official Review · Reviewer_S3Xf · 2023-10-26

**Soundness:** 3 good
**Presentation:** 3 good
**Contribution:** 2 fair
**Rating:** 5
**Confidence:** 4

**Summary:**

The paper discusses the problem of detecting out-of-distribution (OOD) samples in long-tail learning, where models struggle to distinguish tail-class in-distribution samples from OOD samples. The authors introduce a method called Representation Norm Amplification (RNA) to address this challenge by decoupling OOD detection and in-distribution classification. RNA uses the norm of the representation as a new dimension for OOD detection and develops a training method that generates a noticeable discrepancy in the representation norm between ID and OOD data. Experimental results show that RNA outperforms state-of-the-art methods in both OOD detection and classification on CIFAR10-LT, CIFAR100-LT, and ImageNet-LT datasets. The main contributions of the paper are the introduction of RNA as a novel training method and the evaluation of RNA on diverse OOD detection benchmarks.

Contributions:
1. RNA decouples OOD detection and in-distribution classification, different from existing methods struggling to distinguish tail-class in-distribution samples from OOD samples.
2. RNA uses the norm of the representation as a new dimension for OOD detection, which is a noticeable discrepancy in the representation norm between ID and OOD data.
3. Experimental results show that RNA outperforms state-of-the-art methods in OOD detection and classification.

**Strengths:**

Strengths
1. Originality: The proposed method, Representation Norm Amplification (RNA), introduces a new dimension, the norm of representation vectors, for OOD detection. This approach decouples the OOD detection problem from the long-tailed recognition problem, allowing for the simultaneous achievement of both goals without compromising each other.
2. Quality: The results presented in Tables 6, 7, and 8 demonstrate the superior performance of RNA compared to other baseline approaches across various OOD test sets. RNA achieves the best results for all six OOD test sets when trained on CIFAR10-LT and achieves the best results on four OOD test sets as well as on average when trained on CIFAR100-LT.
3.  Clarity: The information provided in the tables is clear and concise, allowing for easy comparison of the performance of different methods. The description of the proposed method, RNA, is also clear and provides a good understanding of how it addresses the limitations of previous approaches.
4.  Significance: The ability to simultaneously achieve high OOD detection performance and accurate long-tailed recognition is significant in various applications, especially in scenarios where both goals are crucial. The proposed RNA method offers a promising solution to this challenge and outperforms other baseline approaches in terms of both OOD detection metrics and classification accuracy.

**Weaknesses:**

1. Reliance on an auxiliary OOD dataset: The proposed RNA method relies on the availability of an auxiliary OOD dataset for effective performance. While the paper mentions the possibility of replacing the auxiliary OOD dataset with augmented training data, it does not provide a thorough exploration of this alternative or compare its performance with the original approach. Further investigation into alternative data augmentation techniques or methods that reduce the reliance on auxiliary OOD datasets would strengthen the proposed method's practicality and generalizability.
2. The study lacks a comprehensive comparison of model performance under different imbalance rates, including balanced datasets, limiting its applicability and understanding of class imbalance effects.

**Questions:**

1. The paper mentions that the proposed RNA method amplifies the norms of only ID representations while indirectly reducing the representation norms of OOD data by updating the BN statistics using both ID and OOD data. It would be beneficial to provide more details on the rationale behind this approach and how it effectively generates a noticeable discrepancy in the representation norm between ID and OOD data. Additionally, It is better to provide theoretical derivation and discuss the potential impact of this approach on the overall training dynamics and convergence behavior would provide further insights.
2. The paper mentions that the proposed RNA method addresses the overconfidence problem of OOD samples and the underconfidence problem of tail-class ID samples. It would be helpful to provide more insights into how the proposed method achieves this and how it compares to other techniques, such as ODIN (Liang et al., 2018) and LogitNorm (Wei et al., 2022), which also aim to mitigate overconfidence issues. Providing a more detailed analysis and comparison of these techniques would enhance the understanding of the proposed method's effectiveness.
3. The robustness of the RNA method to diverse auxiliary datasets remains unclear. Exploring alternative auxiliary datasets holds the potential for improving results, necessitating further investigation into their impact on the RNA method's performance.

---

> ### Author Response · Authors · 2023-11-18
> **Response to Reviewer S3Xf (1/3)**
>
> Thank you for offering valuable feedback. We genuinely appreciate the constructive insights provided and hope to address each of your comment.
>
> **1. Reliance on an auxiliary OOD set**
> > (W1) The proposed RNA method relies on the availability of an auxiliary OOD dataset for effective performance. ... Further investigation into alternative data augmentation techniques or methods that reduce the reliance on auxiliary OOD datasets would strengthen the proposed method's practicality and generalizability.
>
> Our proposed method inherently relies on auxiliary OOD sets. In an effort to mitigate this dependency on auxiliary OOD sets, we explored various techniques for generating auxiliary data samples. These techniques included applying strong augmentations to training samples, such as strong color jittering, inverting, posterizing, rotating, solarizing, and jigsaw puzzles. Additionally, we experimented with gaussian noise samples as auxiliary data. Unfortunately, these types of auxiliary data were ineffective for training models for OOD detection. Following Chen et al. (2021)[g], we found that resizing and cropping the small patches of training samples was moderately effective as OOD data, as reported in Table 4 in the original paper. This strategy achieved a AUROC performance gain of 10.89% over the MSP baseline, and an 8% gain over the case of not utilizing any auxiliary OOD samples but applying RNA loss. Although this outcome still falls short of matching the performance gains achieved by our original method, it indicates the potential for replacing auxiliary OOD samples with augmented training samples in applying our method. For future directions, the development of a novel method for generating auxiliary OOD samples will be crucial in reducing the reliance on OOD auxiliary sets. For example, methods like Grad-CAM, which identify crucial pixels for image classification, can be helpful. Adversarial perturbation or masking techniques can also be employed on these identified pixels to generate auxiliary OOD data with erased semantic information for ID classification.
>
> [g] Chen et al., ATOM: Robustifying Out-of-distribution Detection Using Outlier Mining, ECML 2021.
>
>
> **2. Imbalance ratios**
> > (W2) The study lacks a comprehensive comparison of model performance under different imbalance rates, including balanced datasets, limiting its applicability and understanding of class imbalance effects.
>
> We performed experiments across various imbalance ratio values and reported the results in Appendix E.3. Please refer to Table 10. In particular, we reported the AUROC and classification accuracy (ACC) metrics for the models trained on CIFAR10/100-LT with different imbalance ratios, including the balanced case $\rho=1$. When trained on CIFAR10-LT, our proposed method consistently outperforms OE in terms of both AUROC and ACC across all the imbalance ratios of 1, 10, 50, 100, and 1,000, including the task with a balanced training set. When trained on CIFAR100-LT, RNA improves the accuracy of the model across all the imbalance ratios, but it exhibits relatively lower AUROC than OE when the imbalance ratio is as low as 1 or 10.

---

> ### Author Response · Authors · 2023-11-18
> **Response to Reviewer S3Xf (2/3)**
>
> **3. Rationale behind RNA and the training dynamics**
> > (Q1)The paper mentions that the proposed RNA method amplifies the norms of only ID representations while indirectly reducing the representation norms of OOD data by updating the BN statistics using both ID and OOD data. It would be beneficial to provide more details on the rationale behind this approach and how it effectively generates a noticeable discrepancy in the representation norm between ID and OOD data. Additionally, It is better to provide theoretical derivation and discuss the potential impact of this approach on the overall training dynamics...
>
> In Appendix E.8 in the revised paper, Figure 4 (a,b) depicts the changes of training losses (LA loss $(\mathcal{L}_ {\text{LA}})$ and RNA loss $(\mathcal{L}_ {\text{RNA}})$ over the training, and Figure 4 (c,d) illustrates the dynamics of representation norms of training ID, test ID, training OOD, and test OOD data. These figures demonstrate that the training effectively works to widen the gap between ID and OOD representation norms.
> The rationale behind these outcomes lies in the interaction of the last BN layer, ReLU layer, and the RNA loss. Since the BN layer subtracts the estimated sample mean from input vectors, the elements of the subtracted vectors at each coordinate are  approximately half positive and half negative. Upon passing through the subsequent ReLU layer, approximately half of them are activated, and the rest are deactivated.
> During training with RNA loss, among the input vectors of the last BN layer, ID vectors are enlarged while OOD vectors are not. Thus, as the training proceeds, the following inequalities emerge:
>
> $ f_ {\text{before BN}}(x_ {\text{OOD}})_ k < \mu_ {\text{BN}} < f_ {\text{before BN}}(x_ {\text{ID}})_ k $
>
> for many $k$’s, where $f_ {\text{before BN}}(x)$ represents the latent vector right before the last BN layer, and $f_ {\text{before BN}}(x)_ k$ represents the $k$-th coordinate of $f_ {\text{before BN}}$.
> As a result, passing through the last BN-ReLU layer, many coordinates are activated and enlarged for ID representations, while those are deactivated for of OOD representations. This difference of activation ratios is clearly illustrated in Figure 2(b) in the original paper. The resulting gap in activation ratios directly translates to a gap in representation norms. Consequently, we can leverage the representation norms for effective OOD detection.
>
> We also present the histograms of representation norms for models trained with LA and RNA to visualize the impact of RNA loss on the representation norm distribution. Figure 5 in Appendix E.9 displays the histograms of representation norms of ID test data and OOD test data  on LA/RNA-trained models with CIFAR10-LT and CIFAR100-LT datasets. The evident gap in representation norms between ID test set and OOD test set is shown in the figure for RNA-trained models. This gap in the distributions of representation norms of ID and OOD data enables effective OOD detection using representation norms.
>
>
> **4. Convergence of RNA training**
> > (Q1)...convergence behavior would provide further insights.
>
> The convergence of our designed RNA loss is a pivotal issue. The RNA loss is formulated to train the model to increase the representation norm of ID data, resulting in an increase in the magnitude of training loss(Figure 4(a)(b)). However, it is essential to note that the magnitude of gradient of RNA loss is inversely proportional to the representation norm (see the equations below), so the gradient norm decreases as the training progresses. Additionally, the learning rate is scheduled to decrease over the course of training epochs. In conclusion, RNA training loss demonstrates stable convergence, as illustrated in Figure 4.
>
> $\mathcal{L}_ {\text{RNA}} = - \frac{1}{B} \sum^{B}_ {i=1}\log(1+\|h_ {\phi}(f_ {\theta}(x_ i))\|)$
>
> $h_\phi(f_\theta(x_i)) := h_i$
>
> $\dfrac{\partial \mathcal{L}_{\text{RNA}}}{\partial h_i}=-\dfrac{1}{B(1+\|h_i\|)}\dfrac{h_i}{\|h_i\|}$
>
> $\|\dfrac{\partial \mathcal{L}_{\text{RNA}}}{\partial h_i}\|=\dfrac{1}{B(1+\|h_i\|)}$

---

> ### Author Response · Authors · 2023-11-18
> **Response to Reviewer S3Xf (3/3)**
>
> **5. LogitNorm (Wei et al., 2022) and ODIN (Liang et al., 2018)**
> >(Q2) The paper mentions that the proposed RNA method addresses the overconfidence problem of OOD samples and the underconfidence problem of tail-class ID samples. It would be helpful to provide more insights into how the proposed method achieves this and how it compares to other techniques, such as ODIN (Liang et al., 2018) and LogitNorm (Wei et al., 2022), which also aim to mitigate overconfidence issues. Providing a more detailed analysis and comparison of these techniques would enhance the understanding of the proposed method's effectiveness.
>
> LogitNorm (Wei et al., 2022)[h] addresses the phenomenon where the softmax cross-entropy loss can lead to a continuous increase in the magnitudes of logit vectors, which can result in overly confident predictions. To mitigate this issue, LogitNorm applies normalization to logits during training. ODIN (Liang et al., 2018)[i], on the other hand, empolys temperature scaling for softmax scores at test time to mitigate overconfident predictions. In contrast, our approach focuses on creating a distinct gap between the feature norms of ID and OOD data. This gap aims to foster confident predictions for ID data, while ensuring controlled confidence levels for OOD data. As such, our technique is very different from LogitNorm or ODIN in its underlying objectives.
>
> We further evaluated the LogitNorm (LN) for five different variants: 1) the original LN (normalizing ID data), 2) LN+OE (normalizing aux. OOD samples), 3) LN+OE (normalizing all samples), 4) LN+LA (normalizing ID), and 5) LN+LA+OE (normalizing all samples) for CIFAR10-LT dataset. Among these five variants, the best OOD detection performance in terms of AUROC was achieved by 3) LN+OE (normalizing all samples), while the best classification performance was achieved by 4) LN+LA (normalizing ID). However, the optimal AUROC achieved by 3) LN+OE falls short by 0.44% and by 3.7% compared to the original OE and our approach (RNA), respectively. Similarly, the best classification accuracy achieved by 4) LN+LA (normalizing ID) is less than ours (RNA) by 0.67%. The fact that 5) LN+LA+OE can achieve neither the best OOD detection nor the best classification shows that this approach still suffers from the trade-offs between the two conflicting goals of long-tail learning and OOD detection. Furthermore, the finding that LN+OE is unable to surpass the original OE indicates that the inherent effect of OE itself suffices to regulate the confidence of OOD data, and additional regularization of logit norms by LN offers limited effect.
>
> |Method|AUROC|AUPR|FPR95|ACC|Many|Medium|Few|
> |---|:---:|:---:|:---:|:---:|:---:|:---:|:---:|
> | 1) LN | 84.74 | 83.61 | 51.26 | 73.58 | 94.43 | 72.78 | 53.93 |
> | 2) LN+OE (OOD normalized) | 82.72 | 81.84 | 54.23 | 68.12 | 92.73 | 66.62 | 45.43 |
> | 3) LN+OE (all normalized) | 89.25 | 86.14 | 32.82 | 70.93 | 92.27 | 71.12 | 49.00 |
> | 4) LN+LA | 85.04 | 84.38 | 50.54 | 77.92 | 94.43 | 74.70 | 66.43 |
> | 5) LN+LA+OE (all normalized) | 85.94 | 84.66 | 46.65 | 76.17 | 92.43 | 71.78 | 65.37 |
> | OE | 89.69 | 86.47 | 33.81 | 74.85 | 93.90 | 73.30 | 57.60 |
> | RNA | 92.95 | 92.01 | 28.76 | 78.59 | 93.43 | 74.92 | 68.25 |
>
> [h] Wei et al., Mitigating Neural Network Overconfidence with Logit Normalization, ICML 2022.
>
> [i] Liang, Enhancing The Reliability of Out-of-distribution Image Detection in Neural Networks, ICLR 2018.
>
> **6. Another auxiliary dataset**
> >(Q3) The robustness of the RNA method to diverse auxiliary datasets remains unclear. Exploring alternative auxiliary datasets holds the potential for improving results, necessitating further investigation into their impact on the RNA method's performance.
>
> We conducted experiments with models trained with an alternative auxiliary OOD set, specifically ImageNet-RC when CIFAR10-LT is the ID training set. The detailed results are available in Appendix E.4. Our proposed model outperforms models trained with OE and PASCL in terms of AUROC, FPR95, and total accuracy.

---

### Official Review · Reviewer_P4tp · 2023-11-01

**Soundness:** 3 good
**Presentation:** 3 good
**Contribution:** 3 good
**Rating:** 3
**Confidence:** 4

**Summary:**

This paper proposes a new loss named representation norm amplification to increase the feature norm of ID data during training such that the OOD data could be distinguished from ID data by the feature norm. This method requires an auxiliary OOD dataset to be effective. The loss is applied to the long-tail classification together with the LA loss. Experiments on CIFAR10/100-LT and ImageNet-LT show that the proposed method not only increases the classification accuracy, but also improves the OOD detection performance.

**Strengths:**

- The paper is easy to follow. The trade-off between outlier exposure and logit adjustment is explained to motivate the method.
- Experiments are designed to explain the various design choices.
- The desgin of the RNA loss is new to me, and the observation that the 2-layer projection is better than a no-projection is interesting.

**Weaknesses:**

- The claim "only ID data contributes to the gradient for updating model parameters" (page 2 and page 5) is wrong. OOD data surely affects the model training and their intermediate features appear in the gradient of model parameters. Otherwise you can remove these auxiliary OOD data without affecting the resulting model.
- The motivation only considers the conflict with one of the OOD methods, outlier exposure. However, the motivation of using outlier exposure in the first place is not explained. There are a lot of post-hoc OOD methods, such as Mahalanobis [a], ViM [b], ASH [c], etc that does not requrire training. Does LA conflicts with these post-hoc methods? It is interesting to compare with the performance of LA + a/b/c (to show that exposing OOD in training is necessary) and LA + RNA + a/b/c (to show that the norm is a good OOD scoring function).
- There is only 1 OOD dataset of ImageNet-1K LT, which is inadequate. It would be great to test more OOD datasets such as Texture and OpenImage-O.
- Page 8 "The different trends may be due to the difference in the number of classes of each dataset ... ImageNet-LT generally produce low confidence scores...". It is not obvious to me that the number of classes affect the confidence score. Please provide evidence for this claim.

[a] "Exploring the limits of out-of-distribution detection." NeurIPS 2021.

[b] "ViM: Out-Of-Distribution with Virtual-logit Matching" CVPR 2022.

[c] "Extremely Simple Activation Shaping for Out-of-Distribution Detection" ICLR 2023.

**Questions:**

See weakness. I will raise the score if the questions are reasonably answered.

---

> ### Author Response · Authors · 2023-11-18
> **Response to Reviewer P4tp (1/2)**
>
> Thank you for providing valuable feedback. We appreciate the constructive feedback and carefully address your comments.
>
> **1. Effect of auxiliary OOD in model training**
> > (W1) The claim "only ID data contributes to the gradient for updating model parameters" (page 2 and page 5) is wrong. OOD data surely affects the model training and their intermediate features appear in the gradient of model parameters. Otherwise you can remove these auxiliary OOD data without affecting the resulting model.
>
> As pointed out by the reviewer, both ID and OOD samples influence the training process of RNA. OOD samples are fed forward into the model, leading to updates in the running mean and standard deviation values of BN layers. Subsequently, the updated BN statistics impact the latent vectors of ID data. So, it is accurate to say that OOD samples indirectly influence the updating of the model parameters. Nevertheless, our explicit intention was to ensure that there is no direct gradient update by OOD samples.
> In Sec. 3.1, we demonstrate the gradients of the loss with respect to classifier weights are linear combinations of input sample representations(Eq. 1). Importantly, during RNA-training, there is no gradient term with respect to OOD samples, preventing OOD representations from being involved in the classifier weight updates.
> In summary, while our statements may appear assertive, our underlying intent is to emphasize that there is no direct gradient update by OOD samples, and the gradients do not incorporate OOD representations.
>
> **2. Post-hoc OOD scoring methods**
> > (W2) The motivation only considers the conflict with one of the OOD methods, outlier exposure. However, the motivation of using outlier exposure in the first place is not explained. There are a lot of post-hoc OOD methods, such as Mahalanobis, ViM, ASH, etc that does not requrire training. Does LA conflicts with these post-hoc methods? It is interesting to compare with the performance of LA + a/b/c (to show that exposing OOD in training is necessary) and LA + RNA + a/b/c (to show that the norm is a good OOD scoring function).
>
> The table below reports the OOD detection performance (AUROC) of CE, LA, and RNA-trained models employing various OOD scoring methods, including MSP, Energy, RN, and ASH[f], which is one of the SOTA OOD scoring methods.
> LA+ASH outperforms LA+Energy by 2.47% and 9.24% on CIFAR100-LT and ImageNet-LT, respectively. However, LA+ASH performs worse than RNA+ASH by 7.22% and 6.00% on CIFAR100-LT and ImageNet-LT, respectively, and even worse than RNA+MSP on CIFAR100-LT. This result shows that applying a post-hoc OOD detection method combined with a long-tail learning method may not hurt the ID classification but is not as effective as exposing auxiliary OOD in improving the OOD detection performance.
> We also notice that our RN scoring, which uses the representation norm for OOD detection, outperforms other scoring methods including ASH for RNA-trained models. This implies that the representation norm can encapsulate information useful for OOD detection, and our proposed training method effectively embeds such information about OOD-ness into the representation norm.
>
>
> |  | MSP | Energy | RN | ASH |
> |---|:---:|:---:|:---:|:---:|
> | ID Dataset: CIFAR100-LT |  |  |  |  |
> | CE | 61.39 | 62.65 | 55.67 | 62.78 |
> | LA | 62.86 | 62.46 | 54.20 | 64.93 |
> | RNA | 73.93 | 74.00 | 74.33 | 72.15 |
> | ID Dataset: ImageNet-LT |  |  |  |  |
> | CE | 54.23 | 52.03 | 56.73 | 59.07 |
> | LA | 55.67 | 53.30 | 54.89 | 62.54 |
> | RNA | 61.27 | 61.50 | 75.55 | 68.54 |
>
> [f] Djurisic et al., Extremely Simple Activation Shaping for Out-of-Distribution Detection, ICLR 2023.

---

> > ### Comment · Reviewer_P4tp · 2023-11-22
> > **Response to rebuttal**
> >
> > Thanks for the rebuttal. Looking at the gradient of BN (e.g., https://safakkbilici.github.io/gradient-for-bn/), the average feature map of all samples is involved. So, OOD samples directly affect model parameters.

---

> > > ### Author Response · Authors · 2023-11-22
> > > **Response to Reviewer P4tp**
> > >
> > > We appreciate the reviewer for the comments.
> > >
> > > We reviewed the content in the provided link, specifically examining the gradients involving the BN input samples. However, all the gradients involve the term, $\dfrac{\partial \mathcal{L}}{\partial a_i^r}$, which is the gradient of the loss with respect to the BN output vector of input index r. For r corresponding to OOD samples, this term is consistently zero, as we do not compute the loss with the features of OOD samples.
> > > Furthermore, we do not incorporate the affine transformation in BN layers, setting $\beta=0$ and $\gamma=1$. Consequently there is no need to consider gradients with respect to these terms.
> > > In summary, the gradients do not involve the terms corresponding to OOD samples.

---

> ### Author Response · Authors · 2023-11-18
> **Response to Reviewer P4tp (2/2)**
>
> **3. Test with more OOD datasets**
> > (W3) There is only 1 OOD dataset of ImageNet-1K LT, which is inadequate. It would be great to test more OOD datasets such as Texture and OpenImage-O.
>
> Following the reviewer's suggestion, we expanded our evaluation of OOD detection performance on additional test sets, namely iNaturalist, SUN, Places, Texture, and OpenImage-O. The table below summarizes the AUROC performance values. While our method shows superior performance on iNaturalist, SUN, and Places, it exhibits a performance gap compared to CE, OE, and PASCL models on Texture and OpenImage-O. Notably, on the Texture and OpenImage-O datasets, both OE and PASCL perform worse than the CE baseline model, and our proposed model exhibits a even lower performance. However, when we use MSP score for RNA-trained model on Texture and OpenImage-O, the AUROC values increase by 14.22% and 8.92% on Texture and OpenIamge-O, respectively, compared to using the RN score, and recovers the performance drop or even achieves the best result. Thus, we hypothesize that the primary issue lies in using RN (representation norm) score as the OOD scoring method on Texture dataset and OpenImage-O. We conjecture that this may be attributed from the characteristics of test OOD datasets, and plan to further examine why the representation norm scoring is ineffective for certain OOD test datasets by analyzing the characteristics of the datasets such as inherent overlap or similarity to ID datasets.
>
> | Method | iNaturalist | SUN | Places | Texture | OpenImage-O |
> |---|:---:|:---:|:---:|:---:|:---:|
> | CE | 74.81 | 67.32 | 66.85 | **63.65** | 64.33 |
> | OE | 72.28 | 70.25 | 70.80 | 61.30 | 62.59 |
> | PASCL | 72.63 | 71.15 | 71.57 | 62.45 | 63.25 |
> | RNA (ours) | **81.28** | **73.88** | **74.33** | 45.97 | 57.17 |
> | RNA/MSP | 76.70 | 70.35 | 70.42 | 61.81 | **66.09** |
>
> **4. Confidence trends along the number of classes in the dataset**
> > (W4) Page 8 "The different trends may be due to the difference in the number of classes of each dataset ... ImageNet-LT generally produce low confidence scores...". It is not obvious to me that the number of classes affect the confidence score. Please provide evidence for this claim.
>
> The table below displays the mean confidence values for the training, test, and OOD test sets across three datasets: CIFAR10-LT, CIFAR100-LT, and ImageNet-LT. The models are trained on each dataset using LA loss. Empirically, we observe that as the number of classes increases, the confidence values tend to get lower for both ID and OOD sets.
>
> |  | CIFAR10-LT | CIFAR100-LT | ImageNet-LT |
> |---|:---:|:---:|:---:|
> | # classes | 10 | 100 | 1000 |
> | ID Train | 99.51 | 99.38 | 71.22 |
> | ID Test | 92.44 | 63.83 | 48.99 |
> | OOD (SVHN) | 85.00 | 53.70 | 42.52 |

---

### Official Review · Reviewer_MX6w · 2023-11-03

**Soundness:** 3 good
**Presentation:** 3 good
**Contribution:** 2 fair
**Rating:** 5
**Confidence:** 4

**Summary:**

This paper proposes to amplify the representation norm for detecting OOD samples. The OOD samples are used in training; and the method is specifically applied in the long-tailed classification scenario. The paper claims that the tail data is easily overlapped with the OOD samples. Using representation norm as the OOD detection signal can help to decouple the influence of the training with OOD samples on tail data classification.

**Strengths:**

- The motivation is reasonable. It is important to handle the OOD detection issue with long-tail classification tasks, where the tail data is usually overlapped with the OOD samples.
- The experiments on multiple datasets and various ablation studies are conducted.

**Weaknesses:**

- The idea of using representation norm for OOD detection is not new. And the discussion with related works is missing, such as:

> Park, Jaewoo, Jacky Chen Long Chai, Jaeho Yoon, and Andrew Beng Jin Teoh. "Understanding the Feature Norm for Out-of-Distribution Detection." In Proceedings of the IEEE/CVF International Conference on Computer Vision, pp. 1557-1567. 2023.

- The proposed idea is actually general for OOD detection, especially the setting with OOD samples available in training (OE). It is not specifically designed for long-tail data, although maybe its benefits can be more obvious in long-tail classification settings.

- The experiments can be improved. The datasets and settings used in experiments can be improved. Please check PASCL paper for a reference.

Why the reported performances of the compared methods (such as PASCL) are lower than in the original paper? Please clarify the experimental settings.

**Questions:**

- Please clarify the questions in weaknesses.

- Why the reported performances of the compared methods (such as PASCL) are lower than in the original paper? Please clarify the experimental settings.

---

> ### Author Response · Authors · 2023-11-18
> **Reponse to Reviewer MX6w (1/2)**
>
> Thank you for your constructive feedback. We hope our answers will clear up your concerns.
>
> **1. Using representation norms for OOD detection**
> > (W1) The idea of using representation norm for OOD detection is not new. And the discussion with related works is missing: Park et al. (2023)[d]
>
> As noted by the reviewer, some of the previous works utilize feature or logit norms during training [a,c] or as OOD scores [b,d]. We also utilize feature norms as a primary criteria for OOD detection. However, the main novelty of our work lies in the way we expose auxiliary OOD data during training to generate a gap between the representation norms of ID and OOD data. This gap is established without significantly perturbing the training process for ID classification. We reviewed some of these works in Sec. 5 Related Works and Appendix A, and will add some of the recents work such as [d] in the revised version.
>
> In detail, our work shares a similar perspective with [a,b], where the tendency for larger feature/logit norms in ID data compared to OOD data is recognized. In [a], this tendency is further enforced by introducing a regularizer that increases feature norms for ID data and decreases feature norms for auxiliary OOD data during training. However, our additional ablation study, as shown in Appendix E.4, demonstrates that explicitly employing a loss to reduce the norms of OOD samples during training yields inferior performance in both OOD detection and classification when compared to our RNA loss. This demonstrates the effectiveness of utilizing OOD data not for gradient feedback but purely for the purpose of regularizing the forwarding pass.
>
> Park et al. (2023) [d] proposes a post-hoc OOD scoring method, negative-aware norm (NAN), which captures both the activation and deactivation tendencies of hidden layer neurons to distinguish ID and OOD data. On the other hand, our method focuses on effectively amplifying such a gap during training (without perturbing the long-tail classification) through the RNA loss.
>
> [a] Reducing Network Agnostophobia, Dhamija et al., NeruIPS 2018.
>
> [b] OPEN-SET RECOGNITION: A GOOD CLOSED-SET CLASSIFIER IS ALL YOU NEED, Vaze et al., ICLR 2022.
>
> [c] Mitigating Neural Network Overconfidence with Logit Normalization, Wei et al., 2022.
>
> [d] Understanding the Feature Norm for Out-of-Distribution Detection, Park et al, ICCV 2023.

---

> ### Author Response · Authors · 2023-11-18
> **Reponse to Reviewer MX6w (2/2)**
>
> **2. Our proposed method and LT-OOD task**
> > (W2) The proposed idea is actually general for OOD detection, especially the setting with OOD samples available in training (OE). It is not specifically designed for long-tail data, although maybe its benefits can be more obvious in long-tail classification settings.
>
> As noted by the reviewer, our proposed method is applicable not only to a long-tailed dataset but also to a balanced set. In Appendix Sec. E.3, we reported the AUROC and classification accuracy (ACC) metrics for the models trained on CIFAR10/100-LT with different imbalance ratios, including the balanced case $\rho=1$. When trained on CIFAR10-LT, our proposed method consistently outperforms OE in terms of both AUROC and ACC across all the imbalance ratios of 1, 10, 50, 100, and 1,000, including the task with a balanced training set.
>
> At the same time, we'd like to emphasize that our primary focus is on addressing challenges that arise when OOD detection methods are integrated into long-tail learning settings. As outlined in Sec. 2 and 3.1, when considering the long-tail learning and OOD detection simultaneously, unexpected issues may emerge, such as conflicts in controlling the confidence of rare (tail-class or auxiliary OOD) samples, or the gradients of classifier weights for tail classes being dominated by OOD data. This is the core problem we aim to resolve, and it is the motivation behind introducing our RNA training method for LT-OOD tasks. While our method is versatile enough to be applied in balanced settings, its design is specifically geared towards mitigating challenges of OOD detection in long-tail learning scenarios.
>
> **3. PASCL implementation issue**
> > (W3, Q2) Why the reported performances of the compared methods (such as PASCL) are lower than in the original paper? Please clarify the experimental settings.
>
> We acknowledge the concern regarding the observed performance discrepancy of PASCL [e] in our paper compared to the values reported in the original paper. While we recognize that there is a performance difference, the variation is not substantial. Our proposed method still showcases state-of-the-art performance compared to the metrics reported in the original paper.
> The PASCL performance values in our implementation are derived using the code provided by the authors of the original paper. However, it should be noted that we had to tune the hyperparameters. The publicly available code, when used with the hyperparameters specified in the original paper, did not yield performance levels comparable to those reported in the original paper. So, we further finetuned the hyperparameters to report the best possible performance for PASCL in our experiment. Below we summarize the numbers reported in the original paper and reproduced in our paper. Our proposed method still achieves competitive performance compared to the state-of-the-art results reported in the original paper as summarized below.
>
>
>
> | Method | AUROC | AUPR | FPR95 | ACC |
> |---|:---:|:---:|:---:|:---:|
> | ID Dataset: CIFAR10-LT   |
> | PASCL* | 90.99±0.19 | 89.24±0.34 | 33.36±0.79 | 77.08±1.01 |
> | PASCL** | 90.58±0.29 | 88.44±0.54 | 33.79±0.90 | 76.39±0.56 |
> | RNA (ours) | **92.95±0.13** | **92.01±0.21** |  **28.76±0.53** | **78.59±0.25** |
> | ID Dataset: CIFAR100-LT |  |  |  |  |
> | PASCL* | 73.32±0.32 | 67.18±0.10 | 67.44±0.58 | 43.10±0.47 |
> | PASCL** | 73.14±0.37 |  66.77±0.50 | 67.36±0.46 | 43.44±0.34 |
> | RNA (ours) | **74.33±0.53** | **70.25±0.49** | **68.26±0.94** | **44.39±0.14** |
> | ID Dataset: ImageNet-LT |  |  |  |  |
> | PASCL* | 68.00 | 70.15 |  87.53 | 45.49 |
> | PASCL** | 68.17±0.24 | 70.26±0.31 | 87.62±0.32 | 44.97±1.01 |
> | RNA (ours) | **75.55±0.23** | **74.60±0.27** | **78.16±0.81** | **47.81±0.31** |
>
> *: reported in the original paper
> **: reproduced
>
>
>
> [e] Wang et al., Partial and Asymmetric Contrastive Learning for Out-of-Distribution Detection in Long-Tailed Recognition, ICML 2022.